# Degrees of Freedom for Linear Attention: Distilling Softmax Attention with Optimal Feature Efficiency

**Naoki Nishikawa**
The University of Tokyo, RIKEN AIP
nishikawa-naoki259@g.ecc.u-tokyo.ac.jp

**Rei Higuchi**
The University of Tokyo, RIKEN AIP
higuchi-rei714@g.ecc.u-tokyo.ac.jp

**Taiji Suzuki**
The University of Tokyo, RIKEN AIP
taiji@mist.i.u-tokyo.ac.jp

## Abstract

Linear attention has attracted interest as a computationally efficient approximation to softmax attention, especially for long sequences. Recent studies have explored distilling softmax attention in pre-trained Transformers into linear attention. However, a critical challenge remains: *how to choose the feature dimension that governs the approximation quality*. Existing methods fix this dimension uniformly across all attention layers, overlooking the diverse roles and complexities of them. In this paper, we propose a principled method to automatically determine the feature dimension in linear attention using the concept of statistical *degrees of freedom*, which represent the effective dimensionality of the inputs. We provide a theoretical bound on the approximation error and show that the dimension chosen by our method achieves smaller errors under a fixed computational budget. Furthermore, we introduce an efficient layerwise training strategy to learn nonlinear features tailored to each layer. Experiments on multiple pre-trained transformers demonstrate that our method improves the performance of distilled models compared to baselines without increasing the inference cost. Our findings also provide insight into how the complexity of the attention mechanism evolves across layers.

## 1 Introduction

Transformers have become the standard for sequence modeling across diverse domains such as natural language processing (Vaswani et al., 2017), computer vision (Dosovitskiy, 2020), and speech processing (Dong et al., 2018). A key factor in their success is the attention mechanism, which effectively aggregates the information from input tokens. However, the standard softmax attention requires computing pairwise interactions between all tokens in a sequence, resulting in quadratic time and memory complexity with respect to sequence length. This scalability issue poses significant challenges for large-scale applications.

To address this limitation, numerous efforts have been made to design more efficient alternatives. One prominent approach is *linear attention*, which approximates softmax attention by replacing the kernel between queries and keys with an inner product of finite dimensional features. This reduces both time and memory complexity to linear in the sequence length, enabling scalable inference. The idea was initially proposed by Katharopoulos et al. (2020), and has since been extended in subsequent works (Peng et al., 2021; Choromanski et al., 2021; Qin et al., 2022). Recent architectures based on state space models (SSMs), such as Mamba (Gu and Dao, 2024), are also closely related to linear attention (Dao and Gu, 2024; Wang et al., 2024; Sieber et al., 2024; Han et al., 2024).

39th Conference on Neural Information Processing Systems (NeurIPS 2025).

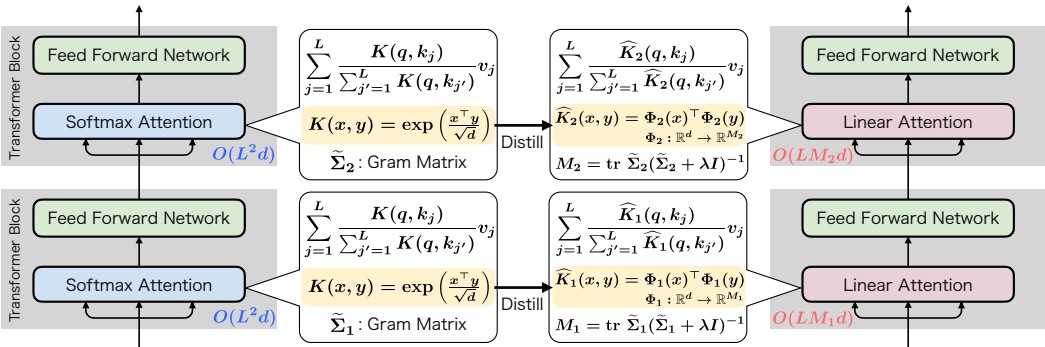

Figure 1: An overview of our method. We treat $\exp\left(x^\top y/\sqrt{d}\right)$, which appears in softmax attention, as a kernel $K(x,y)$, and perform distillation by approximating it with the inner product of nonlinear features. The required feature dimension to achieve a certain error varies depending on the input distribution, and this can be calculated using degrees of freedom derived from the Gram matrix. When the error level $\lambda$ is specified, our method can automatically select the feature dimensions for linear attention, resulting in different feature dimensions for each layer.

Since linear attention is an approximation of softmax attention, one can directly distill the pre-trained softmax attention into linear attention. Even though prior studies (Chen et al., 2024; Wang et al., 2024) have achieved some success in this approach, a critical challenge remains unresolved:

*How should we choose the feature dimension that governs the approximation quality?*

Most existing works pre-determine the feature dimensions and fix them across all layers, which ignores the diverse functional roles and input distributions of different attention layers. As observed in prior studies (Arora et al., 2018; Ravichandran et al., 2019; Suzuki et al., 2020; Massaroli et al., 2024), the complexity of the roles played by each layer varies greatly. It is thus reasonable to expect that selecting the feature dimension adaptively, in accordance with each layer's complexity, can improve the performance of the distilled model.

In this paper, we propose a principled approach to determining the feature dimension in linear attention. We begin by theoretically deriving a bound on the approximation error and show that the number of features required to approximate the original attention kernel is governed by the statistical *degrees of freedom (DoF)*. Based on this insight, we develop a method to automatically select the feature dimension for each layer by estimating its DoF. Because the DoF captures the effective dimensionality of the input distribution, our method enables a data-adaptive and layer-specific allocation of feature dimensions.

Although our theoretical analysis identifies the optimal feature distribution, this distribution is generally intractable to compute. To obtain linear attention that achieves optimal approximation accuracy, we introduce a training strategy for learning nonlinear feature maps. Instead of relying on costly end-to-end training with the pre-training objective, we propose to train each layer independently. This *layerwise training* approach significantly reduces computational cost while allowing each attention layer to learn features tailored to its input distribution.

We validate our approach through extensive experiments on GPT-2 and Pythia-1B. Our results show that (1) the optimal feature dimension varies substantially across layers, (2) our method improves the accuracy of distilled models over fixed-dimension baselines, and (3) our layerwise feature learning yields competitive performance with significantly reduced training cost.

Our contributions can be summarized as follows:

- We propose a principled method for *automatically selecting the feature dimension* of linear attention based on the statistical degrees of freedom, which reflect the complexity of each attention layer. We provide theoretical guarantees on the approximation error under this selection scheme.
- We introduce an efficient *layerwise training* strategy to learn nonlinear features tailored to each layer, which substantially reduces the computational cost compared to end-to-end training.
- We empirically demonstrate that selecting feature dimensions by the proposed method improves performance of the distilled model and yields results comparable to the original models. The experimental results also offer insight into *how attention complexity varies across layers*.

**Other related works.** Some studies aim to distill the attention mechanism in pre-trained Transformers into models based on linear attention. Chen et al. (2024) propose a method to distill softmax attention into linear attention by using a quasi Monte Carlo method, which is more accurate than standard Monte Carlo sampling. Wang et al. (2024) explore a distillation from transformers to Mamba taking into account the similarities between Mamba and linear attention. Furthermore, Bick et al. (2024) and Ralambomihanta et al. (2024) propose methods for distilling Transformers into SSM-based models, and Kasai et al. (2021) develop a method to distill Transformers into RNN-based models. In all of them, methods for selecting feature dimensions based on the complexity of the kernels have not been investigated.

Among numerous studies on the distillation of sequence models, including Transformers (e.g., Wang et al. (2020b), Yang et al. (2021)), Massaroli et al. (2024) and Sakamoto and Sato (2024) are particularly relevant to our research. These studies focus on distilling SSMs into smaller SSMs, and address the selection of state dimensions in the distilled models. However, their methods are highly dependent on the properties of SSMs and cannot be directly applied to the distillation from softmax attention to linear attention.

Wang et al. (2020a) proposes a method to make softmax attention more efficient, and focus on the fact that attention matrices tend to be low-rank. They propose compressing the keys and the values from $L \times d$ to $L' \times d$ $(L' \ll L)$, where $L$ is the sequence length and $d$ is the head size. Our method is similar in that it focuses on the effective dimension calculated from the Gram matrix of the attention kernel. However, we use it for dimension selection in linear attention instead of compressing the keys and values. Furthermore, our method significantly differs from theirs in that the feature dimension is automatically selected using data and takes different values for each layer.

**Notations.** Let $\mu$ be a probability measure on $\mathbb{R}^d$. We define $L_2(\mu)$ as the space of functions $f : \mathbb{R}^d \to \mathbb{R}$ such that $\int f(z)^2 \mu(dz) < \infty$. We denote $\int f(z)\mu(dz)$ as $\mathbb{E}_{z \sim \mu}[f(z)]$ and $\int f(z)g(z)\mu(dz)$ as $\langle f, g \rangle_{L_2(\mu)}$. For probability measures $\mu_1$ and $\mu_2$ on $\mathbb{R}^d$, we denote the product measure as $\mu_1 \otimes \mu_2$. For a bounded linear operator $A : X \to Y$, we denote the operator norm as $\|A\|_{\mathrm{op}}$. For $m \in \mathbb{N}$, we define $[m] := \{1, 2, \ldots, m\}$.

## 2 Attention Mechanism and its Linearization

In this section, we first introduce the regular attention mechanism, and then explain linear attention, which is an approximation of the regular one. For simplicity, in this section, we explain only single-head cases, but the description can be easily extended to multi-head cases as well. Moreover, we focus on unidirectional cases, which are commonly used in language models.

**Attention mechanism.** Let $[x_1, x_2, \ldots, x_L]$ be a sequence of $L$ vectors (called *tokens*), where $x_i \in \mathbb{R}^d$ is the $i$-th vector. The *attention mechanism* computes a new sequence $[y_1, y_2, \ldots, y_L]$ by taking a weighted sum of the projected tokens as follows:

$$y_i = \sum_{j=1}^{i} \frac{K(q_i, k_j)}{\sum_{j'=1}^{i} K(q_i, k_{j'})} v_j, \quad K(x, y) := \exp\left(x^\top y / \sqrt{d}\right), \tag{1}$$

where $k_i := W^K x_i, q_i := W^Q x_i, v_i := W^V x_i$, which are called the *key*, *query*, and *value*, respectively, and $W^K, W^Q, W^V \in \mathbb{R}^{d \times d}$ are the learnable parameters. The kernel $K(x, y)$ is referred to as the *attention kernel*.

The time and memory complexity of the attention mechanism is $\mathcal{O}(L^2 d)$ and $\mathcal{O}(L^2 + Ld)$, respectively. Since we need to compute the inner products between every pair of keys and queries, the cost increases quadratically with the sequence length $L$. Although the attention mechanism is powerful, this becomes a drawback when dealing with the long sequences typical in language and speech.

**Linear attention.** To deal with large computational cost, in *linear attention*, the computation of (1) is simplified by approximating $K(x, y)$ with a inner product of finite dimensional feature maps. Specifically, we suppose that $K$ has a form $K(x, y) = \mathbb{E}_{z \sim \tau}[\phi(x; z)\phi(y; z)]$, where $\tau$ is a probability measure and $\phi(\cdot; z) : \mathbb{R}^d \to \mathbb{R}$ is a feature map. Then, we can approximate $K$ using finite i.i.d. samples $z_1, \ldots, z_M \sim \tau$, i.e., $K(x, y) \approx \frac{1}{M} \sum_{m=1}^{M} \phi(x; z_m)\phi(y; z_m) = \Phi(x)^\top \Phi(y)$, where

$\Phi(x) = \frac{1}{\sqrt{M}}[\phi_1(x), \dots, \phi_M(x)]^\top$. Linear attention approximates the output of (1) by replacing $K(q_i, k_j)$ with $\Phi(q_i)^\top \Phi(k_j)$.

This simplification drastically reduces the computational cost. Specifically, we can compute the outputs in the following form:

$$\hat{y}_i = \sum_{j=1}^{i} \frac{\Phi(k_j)^\top \Phi(q_i)}{\sum_{j'=1}^{i} \Phi(k_{j'})^\top \Phi(q_i)} v_j = \frac{B_i^\top \Phi(q_i)}{A_i^\top \Phi(q_i)}, \quad A_i := \sum_{j=1}^{i} \Phi(k_j), \quad B_i := \sum_{j=1}^{i} \Phi(k_j) v_j^\top.$$

Since $A_i$ and $B_i$ ($i = 1, \dots, L$) can be computed recursively over $i$, the computational complexity with respect to input length $L$ is significantly reduced from quadratic to linear: the time and memory complexities become $\mathcal{O}(LMd)$ and $\mathcal{O}(LM + Ld + Md)$, respectively.

There are multiple options for the feature map $\Phi$. The simplest approach involves using the decomposition $\exp\left(\frac{q^\top k}{\sqrt{d}}\right) = \exp\left(\frac{\|q\|^2}{2\sqrt{d}}\right)\exp\left(\frac{\|k\|^2}{2\sqrt{d}}\right)\exp\left(-\frac{\|q-k\|^2}{2\sqrt{d}}\right)$, and subsequently applying Bochner's theorem to break down the final component, $\exp\left(-\frac{\|q-k\|^2}{2\sqrt{d}}\right)$, into random Fourier features. While this method appears convenient, it suffers from learning instability caused by the appearance of negative values in $\sin$ and $\cos$ functions. To address this issue, Choromanski et al. (2021) propose using the decomposition of $\exp\left(q^\top k/\sqrt{d}\right) = \mathbb{E}_{z \sim \mathcal{N}(0, \mathbb{I}_d)}[\phi(q; z)\,\phi(k; z)]$, where $\phi(x; z) = \exp\left(\frac{z^\top x}{d^{1/4}} - \frac{\|x\|^2}{2\sqrt{d}}\right)$. This feature map is referred to as the *Positive Random Feature (PRF)*. Because its values are consistently positive, it facilitates more stable learning.

## 3   Optimal Distillation from Softmax Attention to Linear Attention

In this section, we develop a theoretical framework and a practical algorithm to select feature dimensions for linear attention, followed by a method to train these features efficiently.

### 3.1   Theoretical Background: Degrees of Freedom for Kernel Approximation

In this subsection, we lay the theoretical groundwork for selecting the optimal feature dimension in linear attention. Our analysis builds on kernel approximation theory and introduces the concept of statistical degrees of freedom (DoF), which quantifies the effective dimensionality required to approximate the attention kernel.

**Finite dimensional approximation of kernel.**   To discuss generally, we consider a continuous positive definite kernel $K : \mathbb{R}^d \times \mathbb{R}^d \to \mathbb{R}$, and suppose that $K$ admits the representation

$$K(x, y) = \mathbb{E}_{z \sim \tau}[\phi(x; z)\phi(y; z)], \tag{2}$$

where $\tau$ is a probability measure on a measurable set $\mathcal{Z}$, and $\phi : \mathbb{R}^d \times \mathcal{Z} \to \mathbb{R}$ is a feature map. We further assume that the function $x \mapsto K(x, x)$ is integrable with respect to $\rho$, and that the map $(x, z) \mapsto \phi(x; z)$ is square-integrable with respect to $\rho \otimes \tau$.

The simplest way to approximate $K$ is to use the empirical kernel $K'$ defined as $K'(x, y) = \frac{1}{M}\sum_{m=1}^{M} \phi(x; z'_m)\phi(y; z'_m)$, where $z'_1, \dots, z'_M$ are i.i.d. samples from $\tau$. According to the strong law of large numbers, $K'$ converges to $K$ almost surely as $M \to \infty$ for any fixed $x$ and $y$. Rahimi and Recht (2007) show that, for any translation-invariant kernel $K$ and compact set $\mathcal{Z} \subset \mathbb{R}^d$, $K'$ uniformly converges to $K$ at a rate of $\mathcal{O}(\log M/\sqrt{M})$.

Rather than focusing on the global error bound over a compact set, we strive to achieve a more accurate approximation by *leveraging the intrinsic structure of the input vectors*. In other words, we assume that the input vectors $x$ and $y$ are generated from a probability measure $\rho$ on $\mathbb{R}^d$, and adjust the approximation scheme utilizing the information of $\rho$. To this end, we consider approximating $K$ using i.i.d. samples $z_1, \dots, z_M$ drawn from a distribution with density $q$ w.r.t. the measure $\tau$:

$$\widehat{K}(x, y) = \frac{1}{M} \sum_{m=1}^{M} \frac{1}{q(z_m)} \phi(x; z_m)\phi(y; z_m) = \Phi(x; z)^\top \Phi(y; z),$$

where $\Phi(x;z) = (M \cdot q(z))^{-1/2}\phi(x;z), z = [z_1,\ldots,z_M]^\top$. The kernel $\widehat{K}$ defines a finite-dimensional Reproducing Kernel Hilbert Space (RKHS) $\widehat{\mathcal{H}}$, which is expected to be an approximation of the RKHS $\mathcal{H}$ defined by $K$. Bach (2017) analyze the approximation error between $\mathcal{H}$ and $\widehat{\mathcal{H}}$, and show that it is characterized by the value $N_{q,\lambda}$ $(\lambda > 0)$ defined as

$$N_{q,\lambda} = \sup_{z \in \mathcal{Z}} \frac{1}{q(z)} \left\langle \phi(\cdot;z), (\Sigma + \lambda I)^{-1}\phi(\cdot;z) \right\rangle_{L_2(\mathrm{d}\rho)},$$

where $\Sigma : L_2(\rho) \to L_2(\rho)$ is an integral operator defined by $(\Sigma f)(x) := \langle K(x,\cdot), f \rangle_{L_2(\rho)}$. Specifically, they provide the following result.

**Proposition 1** (Proposition 1 in Bach (2017))**.** *Suppose that $z_1,\ldots,z_m$ are i.i.d. samples from the distribution with density $q$. Then, for any $\delta \in (0,1)$, if $M \geq 5N_{q,\lambda}\log\frac{16N_{q,\lambda}}{\delta}$, it holds $\frac{1}{M}\sum_{m=1}^M q(z_m)^{-1}\|\phi(\cdot;z_m)\|_{L_2(\mathrm{d}\rho)}^2 \leq \frac{2\operatorname{tr}\Sigma}{\delta}$ and*

$$\sup_{f:\|f\|_{\mathcal{H}}\leq 1} \inf_{\|\beta\|_2^2 \leq \frac{4}{M}} \left\| f - \sum_{m=1}^M \frac{\beta_m}{q(z_m)^{1/2}}\phi(\cdot;z_m) \right\|_{L^2(\rho)}^2 \leq 4\lambda,$$

*with probability at least $1 - \delta$.*

This proposition demonstrates that the necessary number of features $M$ to achieve the approximation error $\lambda$ is determined by $N_{q,\lambda}$, which depends on the density $q$. In other words, by effectively selecting the density $q$, it is possible to achieve an approximation error of $\lambda$ with a smaller $M$.

One of the essential part of the proof of Proposition 1 is to bound the error between the operators $\Sigma$ (corresponding to $K$) and its empirical approximation $\widehat{\Sigma} : L_2(\rho) \to L_2(\rho)$ (corresponding to $\widehat{K}$) defined as $(\widehat{\Sigma}f)(x) := \langle \widehat{K}(x,\cdot), f \rangle_{L_2(\rho)}$. In the proof of Proposition 1, Bach (2017) analyze the error between $\Sigma$ and $\widehat{\Sigma}$ as follows.

**Lemma 2** (Bach (2017))**.** *Let $\Delta_\lambda := (\Sigma + \lambda I)^{-1/2}(\Sigma - \hat{\Sigma})(\Sigma + \lambda I)^{-1/2}$. For any $\lambda, t > 0$, it holds*

$$\mathbb{P}[\|\Delta_\lambda\|_{\mathrm{op}} > t] \leq 2N_{q,\lambda}\left(1 + \frac{6}{t^2\log^2\left(1 + \frac{Mt}{N_{q,\lambda}}\right)}\right)\exp\left(-\frac{Mt^2/2}{N_{q,\lambda}\cdot(1+\frac{t}{3})}\right).$$

This lemma provides the upper bound of the largest difference of eigenvalues between $(\Sigma + \lambda I)^{-1/2}\Sigma(\Sigma + \lambda I)^{-1/2}$ and $(\Sigma + \lambda I)^{-1/2}\widehat{\Sigma}(\Sigma + \lambda I)^{-1/2}$, which is controlled by $N_{q,\lambda}$.

**Error bound between $K$ and $\widehat{K}$.** Proposition 1 shows that the elements of $\mathcal{H}$ can be approximated by those of $\widehat{\mathcal{H}}$ in terms of the $L^2(\rho)$ norm. However, for the purpose of distilling softmax attention into linear attention, it is essential to have a *error bound between $K$ and $\widehat{K}$* and, ultimately, to ensure that *the output of the attention mechanism is well-approximated*. To this end, we establish the following theorem using Lemma 2.

**Theorem 3.** *Let $\delta \in (0,1), \lambda > 0$, and $t \in (0,3]$. Suppose that $M \geq \frac{4N_{q,\lambda}}{t}\log\frac{64N_{q,\lambda}}{\delta t}$. Then, the following two items hold:*

 *(i) It holds*
$$\left\| K - \hat{K} \right\|_{L^2(\rho\otimes\rho)}^2 \leq \lambda \cdot t\delta^{-1}C_K^{(1)} + t^2 C_K^{(2)},$$
 *with probability $1 - 2\delta$, where $C_K^{(1)}, C_K^{(2)}$ are constants depending on $K$.*

 *(ii) Let $\alpha \in [0,1/2]$ and $h \in L^2(\rho)$. We define a map $v : \mathbb{R}^d \to \mathbb{R}$ by $v = (\Sigma + \lambda I)^{-\alpha}h$. Then, it holds*
$$\left\| \int v(x)\hat{K}(x,\cdot)\mathrm{d}\rho(x) - \int v(x)K(x,\cdot)\mathrm{d}\rho(x) \right\|_{L^2(\rho)} \leq \sqrt{2}\left(\lambda^{1-\alpha} + C_K^{(3)}\right)\cdot t\|h\|_{L^2(\rho)},$$
 *with probability $1 - \delta$, where $C_K^{(3)}$ is a constant depending on $K$.*

The proof can be found in Appendix A. We emphasize that this theorem can be applied to any positive definite kernel represented as (2). Item (i) provides a bound on the $L^2(\rho \otimes \rho)$ norm of the difference between the kernels $K$ and $\widehat{K}$, indicating that the approximation error is governed by $N_{q,\lambda}$. This suggests that a careful choice of the sampling density $q$ efficiently reduce the approximation error. Item (ii) bounds the error in kernel-based integration against an arbitrary function $v$. Notably, the integral $\int v(x)K(x,y)\mathrm{d}\rho(x)$ corresponds to the attention-weighted sum in (1)[1]. Thus, this result guarantees that the required feature dimension $M$ for accurately approximating attention outputs is effectively controlled by $N_{q,\lambda}$. Finally, we remark that the space $\{(\Sigma + \lambda I)^{-\alpha} h \mid h \in L^2(\rho)\}$ strictly contains $L^2(\rho)$ when $\alpha > 0$, implying that our bound applies to a broader class of value functions, including those that may not be square-integrable under $\rho$.

**Remark 4.** We compare our theory with two previous studies on the approximation accuracy of linear attention. (i) Choromanski et al. (2021) analyze the approximation quality of PRF-based linear attention in terms of the sup-norm error. They showed that the required number of features $M$ scales as $O(d \log d)$. In contrast, our Theorem 3 bounds the $L^2(\rho \otimes \rho)$-norm to capture the structure of the input distribution. As a result, the approximation error to depend on the DoF rather than the dimension $d$. (ii) Luo et al. (2021) shows exponential dependence on the norm bound $R$ of queries and keys. In our analysis, such dependence is absorbed in constants like $C_K$ and $\|K\|^2_{L^2(\rho \otimes \rho)}$. We do not aim to reduce this exponential dependence, but instead focus on selecting the feature dimension per layer to minimize approximation error under a fixed computational budget.

**Degrees of freedom and their optimality.** Next, we aim to optimally reducing the required value of $N_{q,\lambda}$ for a fixed $\lambda$. Specifically, we consider the density $q_\lambda$ defined as follows:

$$q_\lambda(z) = \frac{1}{\mathrm{tr}\,\Sigma(\Sigma + \lambda I)^{-1}} \left\langle \phi(\cdot;z), (\Sigma + \lambda I)^{-1}\phi(\cdot;z) \right\rangle_{L_2(\rho)}.$$

By setting $q = q_\lambda$, we can see that $N_{q_\lambda,\lambda} = N^*_\lambda := \mathrm{tr}\,\Sigma(\Sigma + \lambda I)^{-1}$. The value $N^*_\lambda$ is called the *degrees of freedom* (DoF).

When we use the i.i.d. samples $z_1, \ldots, z_M$ drawn from a distribution with the density $q_\lambda$, the required number of samples $M$ in Proposition 1 and Theorem 3 is proportional to $N^*_\lambda \log N^*_\lambda$, which is known to be the optimal (Bach (2017), Proposition 3). In particular, the degrees of freedom are always smaller than $N_{1,\lambda}$. This suggests that *the approximation error can be reduced by choosing the distribution $q$ depending on the input distribution $\rho$* compared to the case we obtain the random samples from the distribution $\tau$.

The findings above can be summarized as follows:

> - The feature dimension $M$ should be set proportional to the degrees of freedom $N^*_\lambda = \mathrm{tr}(\Sigma(\Sigma + \lambda I)^{-1})$, up to a logarithmic factor.
> - The degrees of freedom $N^*_\lambda$ depend on the input distribution $\rho$, implying that the optimal feature dimension $M$ varies across layers.
> - Optimal feature efficiency can be achieved by sampling $z_1, \ldots, z_M$ from $q\,d\tau$, rather than from $\tau$. The density $q$ also depends on the input distribution $\rho$.

### 3.2 Feature Dimension Selection via Degrees of Freedom

Given that the degrees of freedom $N^*_\lambda$ determines the minimal required number of features, we now return to the attention kernel $K(x,y) = \exp\left(\frac{x^\top y}{\sqrt{d}}\right)$ and present a practical procedure to estimate it from data and use it to allocate layerwise dimensions. The overview of our method is presented in Algorithm 1.

Since $N^*_\lambda$ is defined as the expectation over the data distribution, it cannot be computed in practice. Therefore, we approximate it using finite samples. Specifically, we define $\widetilde{N}_\lambda$ as an approximation

---

[1]This correspondence becomes exact when $W^K$ is invertible. Letting $\rho' := W^K_\# \rho$ denote the pushforward measure of $\rho$ by $W^K$, the attention-weighted sum of values is represented as $\int v(k)K(k,y)\mathrm{d}\rho'(k)$ with $k = W^K x$, $v(k) = W^V(W^K)^{-1}k$.

---

**Algorithm 1** Selecting the Feature Dimension

---

**Input:** Set $\mathcal{X}$ of $T$ sequences with length $L$, sample size $J \in \mathbb{N}$, tolerance $\lambda > 0$, cost $C \in \mathbb{N}$, original Transformer with $S$ layers and $H$ heads.

Fed the sequences in $\mathcal{X}$ into the pre-trained Transformer and collect the queries $Q_{s,h}$ and keys $K_{s,h}$ of all layers $s \in [S]$ and heads $h \in [H]$.

**for** $s = 1$ **to** $S$ **do**
    **for** $h = 1$ **to** $H$ **do**
        Randomly sample $x_1, \ldots, x_J$ from $Q_{s,h} \cup K_{s,h}$ and compute the Gram matrix $\widetilde{\Sigma}_{s,h} \in \mathbb{R}^{J \times J}$.

        Compute $\widetilde{N}_\lambda^{(s,h)} = \operatorname{tr} \widetilde{\Sigma}_{s,h}(\widetilde{\Sigma}_{s,h} + \lambda I)^{-1}$.
    **end for**
    Obtain $\widetilde{N}_\lambda^{(s)} = \max_{h \in [H]} \widetilde{N}_\lambda^{(s,h)}$.
**end for**
Set $t^{-1} = C \cdot \left( S^{-1} \sum_{s=1}^{S} \widetilde{N}_\lambda^{(s)} \right)^{-1}$.

**Return:** Feature dimension $M_s = \operatorname{round}(t^{-1} \widetilde{N}_\lambda^{(s)})$ for layers $s = 1, \ldots, S$.

---

of $N_\lambda^*$, given by $\widetilde{N}_\lambda = \operatorname{tr} \widetilde{\Sigma}(\widetilde{\Sigma} + \lambda I)^{-1}$, where $\widetilde{\Sigma} : \mathbb{R}^J \to \mathbb{R}^J$ ($J \in \mathbb{N}$) is a gram matrix of $K$, i.e., $\widetilde{\Sigma} = [K(x_i, x_j)]_{i \in [J], j \in [J]}$, which is the approximation of $\Sigma$ based on finite samples $x_1, \ldots, x_J \sim \rho$.

For the attention kernel, the inputs are the queries and keys of the attention layers. Therefore, we compute the approximated degrees of freedom $\widetilde{N}_\lambda$ for each layer as follows:

1. Prepare $T$ sequences of length $L$, and compute queries and keys for all sequences and tokens, resulting in $LT$ queries and $LT$ keys, respectively.

2. From the collected $2LT$ queries and keys, randomly sample $J$ elements $x_1, \ldots, x_J$.

3. Compute approximated degrees of freedom $\widetilde{N}_\lambda$ defined above.

The degrees of freedom obtained through this procedure vary across heads and layers, as shown in Section 4. In typical Transformer implementations, the heads within the same layer are processed using a shared tensor, so it is desirable that the feature dimensions be identical as well. Hence, we define the degrees of freedom $\widetilde{N}_\lambda^{(s)}$ of the $s$-th layer as the maximum degrees of freedom among the heads.

According to the results of Theorem 3 (excluding the logarithm), we set the feature dimension of the $s$-th layer to $M^{(s)} = t^{-1} \widetilde{N}_\lambda^{(s)}$ for some $t > 0$. Then, the computational cost of inference matches that of linear attention with a fixed feature dimension given by $\frac{1}{S} \sum_{s=1}^{S} M^{(s)} = t^{-1} \cdot \frac{1}{S} \sum_{s=1}^{S} \widetilde{N}_\lambda^{(s)}$. In our experiments, we fixed $\lambda$ in advance. Given a computational cost $C$, we determine the constant $t$ such that the inference cost of the model matches the computational cost when we the feature dimension is uniformly set to $C$ across all layers, which leads $t^{-1} = C \cdot \left( \frac{1}{S} \sum_{s=1}^{S} \widetilde{N}_\lambda^{(s)} \right)^{-1}$.

### 3.3 Layerwise Training of Features for Kernel Approximation

Theorem 3 guarantees that we can attain the optimal error between $K$ and $\widehat{K}$ by choosing the density $q$ appropriately, However, it is difficult to obtain such a density in practice. To obtain features $z_i$ and $q(z_i)^{-1}(=: \alpha_i)$ ($i \in [M]$) that approximate the attention kernel as accurately as our theoretical bound, we propose to learn them from the training data.

One possible strategy is to train the features using the same loss function as the one used in the standard pre-training of Transformers. That said, this approach is computationally expensive, as it requires computing gradients across all layers simultaneously. Instead, we propose to train the features in a layerwise manner. Specifically, we consider two types of loss functions: **(i)** $L^2$ **loss** and **(ii) softmax loss**. To describe the details, let us assume that the set of queries $\mathcal{Q}_t = (q_{t,1}, \ldots, q_{t,L})$ and keys $\mathcal{K}_t = (k_{t,1}, \ldots, k_{t,L})$ of $T$ sequences, each of length $L$, are given.

Table 1: The dimension of features selected by our method. The feature dimensions of each layer are determined to match the average cost. For each row, the top three largest dimensions are highlighted in bold, and the three lowest dimensions are emphasized with underlines.

| Model | Cost | 1 | 2 | 3 | 4 | 5 | 6 | 7 | 8 | 9 | 10 | 11 | 12 | 13 | 14 | 15 | 16 |
|---|---|---|---|---|---|---|---|---|---|---|---|---|---|---|---|---|---|
| GPT-2 | 64 | **130** | **182** | 35 | 44 | 42 | 65 | **92** | 33 | 28 | 34 | 46 | 39 | - | - | - | - |
| Pythia-1B | 128 | **277** | 138 | **275** | 132 | 204 | 167 | 115 | 142 | **231** | 64 | 96 | 73 | 40 | 52 | 34 | 9 |

**(i) $L^2$ loss.** This loss function tries to minimize the error between the attention kernel $K$ and its approximation $\widehat{K}$ in terms of the $L^2$ norm. That is, the features $z = (z_1, \ldots, z_M)$ are trained using the loss function defined as $\ell(z) = \frac{1}{LT} \sum_{t \in [T], l \in [L], l' \in [L]} \left( K(q_{t,l}, k_{t,l'}) - \widehat{K}(q_{t,l}, k_{t,l'}; z, \alpha) \right)^2$.

**(ii) Softmax loss.** We train the features to make the attention matrix and its approximation closer in terms of the cross entropy loss. In other words, we train the features $z = (z_1, \ldots, z_M)$ defined as $\ell(z) = -\frac{1}{LT} \sum_{t \in [T], l \in [L], l' \in [L]} p_{l'}(q_{t,l}, \mathcal{K}_t) \log \widehat{p}_{l'}(q_{t,l}, \mathcal{K}_t; z, \alpha)$, where $p_{l'}(q_{t,l}, \mathcal{K}_t) = \frac{\exp(q_{t,l}^\top k_{t,l'})}{\sum_{j=1}^{L} \exp(q_{t,l}^\top k_{t,j})}$, $\widehat{p}_{l'}(q_{t,l}, \mathcal{K}_t; z, \alpha) = \frac{\widehat{K}(q_{t,l}, k_{t,l'}; z, \alpha)}{\sum_{j=1}^{L} \widehat{K}(q_{t,l}, k_{t,j}; z, \alpha)}$.

In general, $L^2$ loss is more efficient than Softmax loss because it does not require additional nonlinear operations such as $\exp$ or normalization. In Section 4, we compare the performance of the two losses (i) and (ii) with the case where the features are directly learned using the pre-training task. Then, we demonstrate that learning with our proposed loss is as effective as training using pre-training loss.

## 4 Experiments

To evaluate the effectiveness of our method, we conduct experiments on two types of pre-trained Transformers: GPT-2 (Radford et al., 2019) and Pythia-1B (Biderman et al., 2023). As non-linear features for the linear attention, we use PRF (Choromanski et al., 2021), which we describe in Section 2. For dimension selection and training of features, we utilize the Wikipedia dataset[2].

### 4.1 Optimal Feature Dimensions *Vary across Layers and Heads*

First, we apply Algorithm 1 to the two pre-trained models and determine the feature dimensions for each layer. We set the cost $C$ to be the same as the head size (64 for GPT-2 and 128 for Pythia-1B), following prior work that sets the feature dimension of each layer to match the head size. Furthermore, we set $\lambda = 2^{-4}$ for GPT-2 and $\lambda = 2^{-8}$ for Pythia-1B. We present the selected feature dimensions in Table 1. We also show the estimated DoF for each head in GPT-2 in Table 2. We summarize the key findings from these results below.

Table 2: DoF with $\lambda = 2^{-8}$ of each head. The row and column represent the layer and head, respectively. The maximum value of each layer is highlighted in bold.

| Layer | 1 | 2 | 3 | 4 | 5 | 6 | 7 | 8 | 9 | 10 | 11 | 12 |
|---|---|---|---|---|---|---|---|---|---|---|---|---|
| 1 | 4.7 | 8.1 | 4.5 | 2.0 | 3.6 | 16.2 | 4.5 | 2.6 | 4.1 | 6.4 | 5.0 | **150.0** |
| 2 | 10.5 | 28.5 | 155.1 | 44.6 | **173.8** | 51.1 | 9.9 | 16.5 | 17.5 | 8.1 | 88.9 | 19.5 |
| 3 | 9.9 | 8.1 | 1.8 | 2.7 | 3.4 | 2.5 | **24.5** | 5.8 | 2.8 | 2.6 | 12.1 | 6.4 |
| 4 | 15.1 | 3.0 | 1.4 | 1.7 | **39.8** | 12.1 | 1.6 | 1.5 | 1.5 | 2.5 | 12.6 | 1.9 |
| 5 | 1.8 | 1.8 | 11.3 | 1.8 | 42.1 | 2.3 | 4.0 | 7.1 | 22.6 | 3.3 | **31.6** | 1.1 |
| 6 | 8.9 | 3.1 | 3.7 | 3.5 | 3.7 | 3.0 | 6.9 | 7.5 | 4.7 | 52.4 | 21.4 | **66.6** |
| 7 | 3.3 | 6.9 | 68.9 | 14.4 | 43.5 | 17.2 | **107.4** | 6.0 | 4.3 | 1.8 | 32.2 | 3.0 |
| 8 | 3.6 | 11.9 | 3.1 | 12.3 | 7.6 | 17.5 | **33.0** | 9.1 | 4.9 | 5.1 | 2.3 | 12.6 |
| 9 | **24.9** | 6.5 | 19.6 | 9.3 | 5.5 | 3.2 | 10.2 | 3.1 | 5.7 | 16.7 | 22.4 | 3.9 |
| 10 | 18.5 | 13.1 | 15.9 | 3.3 | 22.7 | **29.9** | 14.1 | 6.4 | 20.2 | 5.1 | 3.9 | 6.2 |
| 11 | 7.9 | 6.6 | 26.1 | 19.8 | 18.2 | 3.3 | 8.7 | 6.7 | **43.2** | 2.3 | 17.1 | 13.5 |
| 12 | 5.1 | 21.8 | 22.5 | 10.6 | 30.0 | **39.8** | 27.2 | 31.3 | 5.0 | 20.0 | 2.8 | 2.6 |

**The evolution of effective dimensionality across layers.** The result indicates that the effective dimensionality varies significantly across layers. This emphasizes the difference of the complexity of the roles played by each layer. Taking a closer look at the results, we observe that the feature dimensions selected by our method are generally *larger in the early and middle layers*, and *small in the latter layers*. This shows that Transformers engage in complex token interactions from the shallow layers to the middle layers, and relatively simple processing in the later layers. This observation aligns with the insights obtained in previous studies on fully-connected/convolutional neural networks (Arora et al., 2018; Ravichandran et al., 2019; Suzuki et al., 2020). Furthermore, it is worth noting that, between the early layers and the middle layers, there are several layers with relatively small dimensions.

---

[2]The dataset can be accessed through the Hugging Face library: `https://huggingface.co/datasets/legacy-datasets/wikipedia`

Table 3: The results of distillation for GPT-2. *Fix*, *DoF* and *DoF + Clip* in the first column imply the strategies of dimension selection. For all downstream tasks, the evaluation metric is accuracy. The top three highest-performing are highlighted in bold.

| Strategy | Method | PiQA | logiQA | ARC-E | ARC-C | Winogrande | MMLU | WSC | Average |
|---|---|---|---|---|---|---|---|---|---|
| **Original GPT-2** | | **0.5985** | 0.3103 | **0.3325** | 0.3003 | 0.5122 | 0.2789 | **0.6538** | **0.4266** |
| — | DiJiang | 0.5065 | 0.2550 | 0.2113 | 0.2244 | 0.4846 | 0.2639 | 0.4615 | 0.3409 |
| | Performer | 0.5468 | 0.2934 | 0.3039 | 0.2747 | 0.4996 | 0.2517 | 0.5962 | 0.3952 |
| Fix | direct | 0.5832 | **0.3195** | 0.2921 | 0.2995 | **0.5335** | 0.2552 | 0.6154 | 0.4141 |
| | softmax | 0.5718 | 0.2673 | 0.2479 | **0.3029** | 0.5020 | 0.2634 | 0.6154 | 0.3958 |
| | $L^2$ | 0.5822 | **0.3195** | 0.2483 | 0.2773 | 0.5107 | 0.2520 | 0.5962 | 0.3980 |
| DoF | direct | 0.5669 | 0.3011 | **0.3241** | **0.3012** | **0.5280** | 0.2712 | **0.6346** | 0.4182 |
| | softmax | 0.5751 | 0.3026 | 0.3224 | 0.2995 | **0.5328** | 0.2564 | **0.6442** | **0.4190** |
| | $L^2$ | 0.5664 | 0.3088 | 0.2736 | 0.2824 | 0.4972 | 0.2608 | 0.5865 | 0.3965 |
| DoF + Clip | direct | **0.5892** | 0.3164 | 0.3136 | 0.2952 | 0.5075 | **0.2993** | 0.6346 | **0.4223** |
| | softmax | **0.5860** | 0.3026 | **0.3401** | 0.2816 | 0.4996 | **0.2832** | 0.6346 | 0.4182 |
| | $L^2$ | 0.5822 | **0.3164** | 0.2942 | **0.3063** | 0.5091 | **0.2799** | 0.5673 | 0.4079 |

**Comparison of DoF across heads in the same layer.** We additionally observe that the degrees of freedom are largely different across heads as well. Interestingly, *most of the heads have much smaller degrees of freedom than the maximum of each layer*. This indicates that the heads performing complex processing are limited to a few within a single layer.

## 4.2 Dimension Selection with Layerwise Training *Enhances the Performance*

We now evaluate the effectiveness of our method in distillation from softmax attention for two pre-trained models. In these experiments, we examine the following two claims: (i) selecting feature dimensions using our DoF-based method improve performance compared to using fixed dimensions, and (ii) our efficient layerwise training match the performance of full end-to-end training.

We consider three types of dimension selection strategies: *Fix* is the method that sets the feature dimensions to the same value across all layers, which is conventionally used in existing works. We set the feature dimensions in Fix to be the same as the head size, as in prior work (Chen et al., 2024). *DoF* is the method that selects the feature dimensions using Algorithm 1 (i.e., proportional to the maximum DoFs among heads). We also consider *DoF + Clip*, in which we clamp the feature dimensions to be at most the head size. This strategy is expected to maintain the performance of Fix, while reducing the computational cost.

For training the feature maps during distillation, we consider three types of loss functions: the first is *direct* loss, which we refer to the cross-entropy loss for next-token prediction, which provides the best performance in the pre-training task but requires full end-to-end training. The remaining two are *softmax* loss and $L^2$ loss, which are the layerwise losses introduced in Section 3.3, and more efficient and allow independent training per layer. We will show that the layerwise training achieves performance comparable to the direct loss, while significantly reducing training cost.

As comparison baselines, we evaluate against Performer (Choromanski et al., 2021) and DiJiang (Chen et al., 2024). Both methods employ PRF for linear attention but do not incorporate any data-driven dimension selection. DiJiang additionally utilizes quasi Monte Carlo sampling of features. The learnable parameters of DiJiang are trained on the same dataset used in distillation.

To assess downstream performance, we fine-tune all distilled models (as well as the original Transformers) on a suite of seven tasks: PiQA (Bisk et al., 2020), logiQA (Liu et al., 2020), ARC-Easy/Challenge (Clark et al., 2018), Winogrande (Sakaguchi et al., 2019), WSC (Levesque et al., 2011), and MMLU (Hendrycks et al., 2021). Further details on the training settings and datasets are provided in Appendix C.

**Results for GPT-2.** We first show the experimental results for GPT-2 in Table 3. We outline the noteworthy four observations below.

- **Enhanced performance through dimension selection:** Distilled models using feature dimensions selected by Algorithm 1 (DoF) achieve higher or near-equal performance compared to those using fixed dimensions (Fix). In particular, in the average performance across all tasks (see the rightmost column of Table 3), when using direct or softmax loss, the distilled models with DoF significantly outperforms the ones with Fix (e.g., *75 % mitigation of performance degradation*

Table 4: Comparison of the inference time per sequence with 1,000 tokens for different feature dimensions (for GPT-2).

|  | Fix | DoF | DoF + Clip |
|---|---|---|---|
| Speed (sec/sequence) | 15.589 | 15.579 | 14.662 |

Table 5: Comparison of the consuming time per each sample during training for different loss types (for GPT-2).

| Type of loss | direct | softmax | $L^2$ |
|---|---|---|---|
| Speed (ms/sample) | 109.4 | 90.4 | 60.6 |

Table 6: The downstream accuracy for Pythia-1B. We use softmax loss for distillation.

| Method | PiQA | logiQA | ARC-E | ARC-C | Winogrande | MMLU | WSC | Average |
|---|---|---|---|---|---|---|---|---|
| **Original Pythia-1B** | 0.6213 | 0.2995 | 0.3295 | 0.3072 | 0.5146 | 0.2883 | 0.6346 | 0.4279 |
| Fix | 0.5691 | 0.2965 | 0.2849 | 0.2679 | 0.5051 | 0.2903 | 0.5962 | 0.4014 |
| DoF | 0.5860 | 0.3026 | 0.3110 | 0.3106 | 0.4949 | 0.2811 | 0.5962 | 0.4118 |

*from the original model* for softmax loss). This indicates that our method effectively captures the varying complexity of each layer, leading to improved accuracy across downstream tasks.

- **Effectiveness of DoF + Clip for efficiency:** The DoF + Clip strategy, which caps feature dimensions at the head size, achieves comparable or better performance than the fixed-dimension approach. Notably, it also yields faster inference (Table 4), showing that selectively reducing feature dimensions based on DoF makes the model efficient without compromising performance.

- **Stronger results compared to prior baselines:** Our distilled models surpass both Performer and DiJiang, which use fixed-dimension linear attention and (quasi) Monte Carlo feature sampling. Additionally, the average performace our distilled model is comparable to (and sometimes win) the original GPT-2 model. This shows that our data-adaptive, layer-specific dimension selection combined with learned features yields a more accurate approximation of softmax attention.

- **Efficient and effective layerwise feature training:** Using the softmax loss, distilled models perform comparably to those trained end-to-end with the standard pre-training loss (direct). As shown in Table 5, layerwise training is more efficient than direct training. Although the L2 loss yields slightly lower performance than both direct and softmax losses, it still outperforms prior baselines while offering notable training efficiency gains.

**Results for Pythia-1B.** We also carried out experiments on distilling Pythia-1B. In this experiment, the model was distilled using the softmax loss. The results are presented in Table 6. The distilled model with feature dimensions selected by our method performs better than the model with fixed dimension on average, and become comparable to the original Pythia-1B. Indeed, when comparing the average accuracy across all tasks, the performance degradation from the original Pythia-1B (Fix: $-0.0265$, DoF: $-0.0161$) is reduced by 39 %. This result indicates the efficacy of our dimension selection and layerwise training when distilling large models.

## 5   Conclusion

We proposed a principled method for selecting feature dimensions in linear attention, grounded in statistical theory on finite-dimensional kernel approximation, and demonstrated its effectiveness in distilling softmax attention. Our approach adaptively assigns dimensions to each layer based on the statistical degrees of freedom of attention kernel, and trains nonlinear features in a layerwise manner to reduce computation. Experiments on GPT-2 and Pythia-1B show that our method improves performance without increasing inference cost, highlighting the importance of layer-specific design for efficient linear attention models.

**Limitation and Future Work.** While our method assigns feature dimensions adaptively per layer, we use a shared dimension across all heads within a layer to maintain implementation efficiency. Interestingly, our analysis revealed that the DoFs vary significantly even among heads in the same layer. Exploiting this head-level variability—for instance, through head pruning or sparse feature allocation—could further enhance performance and efficiency. In addition, although our experiments focus on moderate-scale language models, applying our approach to larger models and models for other modalities remain an important direction to validate its scalability.

## Acknowledgements

NN was partially supported by JST ACT-X (JPMJAX24CK) and JST BOOST (JPMJBS2418). RH was partially supported by JST CREST (JPMJCR2115). TS was partially supported by JSPS KAKENHI (24K02905) and JST CREST (JPMJCR2015). This research is supported by the National Research Foundation, Singapore and the Ministry of Digital Development and Information under the AI Visiting Professorship Programme (award number AIVP-2024-004). Any opinions, findings and conclusions or recommendations expressed in this material are those of the author(s) and do not reflect the views of National Research Foundation, Singapore and the Ministry of Digital Development and Information.

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

# —— Appendix ——

## A  Proof of Theorem 3

We begin by recalling the definitions of some symbols used in the proof.

- $K : \mathbb{R}^d \times \mathbb{R}^d \to \mathbb{R}$ is the positive definite kernel given by $K(x,y) = \mathbb{E}_{z \sim \tau}[\phi(x;z)\phi(y;z)]$, where $\tau$ is a probability measure on a measurable set $\mathcal{Z}$, and $\phi : \mathbb{R}^d \times \mathcal{Z} \to \mathbb{R}$ is a feature map.

- $\Sigma : L_2(\rho) \to L_2(\rho)$ is an integral operator defined by $(\Sigma f)(x) := \langle K(x,\cdot), f \rangle_{L_2(\rho)}$.

- $\widehat{K} : \mathbb{R}^d \times \mathbb{R}^d \to \mathbb{R}$ is the approximation of $K$, defined as $\widehat{K}(x,y) = \frac{1}{M}\sum_{m=1}^{M} \phi(x;z_m)\phi(y;z_m)$, where $z_1, \ldots, z_M$ are i.i.d. samples drawn from a distribution with density $q$ w.r.t. the measure $\tau$.

- $\widehat{\Sigma} : L_2(\rho) \to L_2(\rho)$ is the empirical approximation of $\Sigma$, defined as $(\widehat{\Sigma} f)(x) := \langle \widehat{K}(x,\cdot), f \rangle_{L_2(\rho)}$.

- $N_{q,\lambda}$ ($\lambda > 0$) is the value defined as $N_{q,\lambda} := \sup_{z \in \mathcal{Z}} \frac{1}{q(z)} \langle \phi(\cdot;z), (\Sigma + \lambda I)^{-1}\phi(\cdot;z) \rangle_{L_2(\mathrm{d}\rho)}$.

We restate Lemma 2, which can be found in the proof of Proposition 1 in Bach (2017).

**Lemma 5.** *Let $\Delta_\lambda := (\Sigma + \lambda I)^{-1/2}(\Sigma - \hat{\Sigma})(\Sigma + \lambda I)^{-1/2}$. For any $\lambda, t > 0$, it holds*

$$\mathbb{P}[\|\Delta_\lambda\|_{\mathrm{op}} > t] \leq 2N_{q,\lambda}\left(1 + \frac{6}{t^2 \log^2(1 + Mt/N_{q,\lambda})}\right) \exp\left(-\frac{Mt^2/2}{N_{q,\lambda}(1 + t/3)}\right).$$

This lemma produces the following high probability bound.

**Lemma 6.** *For any $\lambda, \delta \in (0,1)$, if*

$$M \geq \frac{4N_{q,\lambda}}{t^2} \log \frac{64N_{q,\lambda}}{\delta t^2},$$

*it holds*

$$\|\Delta_\lambda\|_{\mathrm{op}} \leq t, \quad \text{and equivalently,} \quad -t(\Sigma + \lambda I) \preceq \Sigma - \hat{\Sigma} \preceq t(\Sigma + \lambda I)$$

*with probability at least $1 - \delta$.*

*Proof.* We first note that

$$2N_{q,\lambda}\left(1 + \frac{6}{t^2 \log^2(1 + Mt/N_{q,\lambda})}\right) \exp\left(-\frac{Mt^2/2}{N_{q,\lambda}(1 + t/3)}\right)$$

$$\leq \frac{2N_{q,\lambda}}{t^2} \underbrace{\left(t^2 + \frac{6}{\log^2(1 + Mt/N_{q,\lambda})}\right)}_{(a)} \underbrace{\exp\left(-\frac{Mt^2/2}{N_{q,\lambda}(1 + t/3)}\right)}_{(b)}$$

Let us consider

$$M \geq B\frac{N_{q,\lambda}}{t^2} \log \frac{CN_{q,\lambda}}{\delta t^2},$$

for some constants $B, C > 0$ which will be determined later. First, we consider the factor (b). Then, for any $t \in (0,3]$, we have

$$(b) \leq \exp\left(-\frac{B/2}{1 + t/3} \log \frac{CN_{q,\lambda}}{\delta t^2}\right) \leq \left(\frac{\delta t^2}{CN_{q,\lambda}}\right)^{\frac{B/2}{1+t/3}} \leq \left(\frac{\delta t^2}{CN_{q,\lambda}}\right)^{B/b^\dagger},$$

where

$$b^\dagger = \begin{cases} 4 & \text{if } \delta t^2/CN_{q,\lambda} \leq 1, \\ 2 & \text{if } \delta t^2/CN_{q,\lambda} > 1. \end{cases}$$

Next, we consider the factor (a). Let $D \in [3/C, \infty)$ be a constant to be determined later. Then, if $N_{q,\lambda}/t \geq D$, we have

$$\frac{Mt}{N_{q,\lambda}} \geq \frac{B}{t} \log \frac{CN_{q,\lambda}}{\delta t^2} \geq \frac{B}{t} \log \frac{CD}{\delta t} \geq \frac{B}{t} \log \frac{CD}{3} \geq \frac{B}{3} \log \frac{CD}{3},$$

which implies

$$\text{(a)} \leq 9 + \frac{6}{\log^2 \left( 1 + \frac{B}{3} \log \frac{CD}{3} \right)}.$$

On the other hand, if $N_{q,\lambda}/t \leq D$, then $Mt/N_{q,\lambda} \geq 1/D$ since $M \geq 1$, which yields

$$\text{(a)} \leq 9 + \frac{6}{\log^2 \left( 1 + \frac{1}{D} \right)}.$$

Finally, we choose $B, C, D > 0$. If we set $B = b^\dagger$, we obtain the upper bound of the probability as

$$\frac{2N_{q,\lambda}}{t^2} \cdot \text{(a)} \cdot \frac{\delta t^2}{CN_{q,\lambda}} = \frac{2\delta}{C} \cdot \text{(a)}.$$

Moreover, we have

$$\text{(a)} \leq \max \left\{ 9 + \frac{6}{\log^2 \left( 1 + \frac{b^\dagger}{3} \log \frac{CD}{3} \right)}, 9 + \frac{6}{\log^2 \left( 1 + \frac{1}{D} \right)} \right\}$$

$$\leq 9 + \frac{6}{\log^2 \left( \min \left\{ 1 + \frac{2}{3} \log \frac{CD}{3}, 1 + \frac{1}{D} \right\} \right)}. \tag{3}$$

If we set $C = 64, D = 1/2$, then it holds $D \in [3/C, \infty)$, and the right-hand side of (3) is smaller than 32, which implies (a) $< 32$. Thus, if we set

$$M \geq \frac{4N_{q,\lambda}}{t^2} \log \frac{64N_{q,\lambda}}{\delta t^2} \geq \frac{b^\dagger N_{q,\lambda}}{t^2} \log \frac{64N_{q,\lambda}}{\delta t^2},$$

we have $\|\Delta_\lambda\|_{\text{op}} \leq t$ with probability at least $1 - \delta$.

The result $\|\Delta_\lambda\|_{\text{op}} \leq t$ implies that

$$-tI \preceq (\Sigma + \lambda I)^{-1/2}(\Sigma - \hat{\Sigma})(\Sigma + \lambda I)^{-1/2} \preceq tI.$$

From the right-hand side inequality, we have

$$\Sigma - \hat{\Sigma} \preceq t(\Sigma + \lambda I).$$

Indeed, for any $f \in L_2(\mathrm{d}\rho)$, we have

$$\left\langle f, (t(\Sigma + \lambda I) - (\Sigma - \hat{\Sigma}))f \right\rangle_{L_2(\mathrm{d}\rho)}$$

$$= \left\langle f, (\Sigma + \lambda I)^{1/2}(tI - (\Sigma + \lambda I)^{-1/2}(\Sigma - \hat{\Sigma})(\Sigma + \lambda I)^{-1/2})(\Sigma + \lambda I)^{1/2}f \right\rangle_{L_2(\mathrm{d}\rho)}$$

$$= \left\langle (\Sigma + \lambda I)^{1/2}f, (tI - (\Sigma + \lambda I)^{-1/2}(\Sigma - \hat{\Sigma})(\Sigma + \lambda I)^{-1/2})(\Sigma + \lambda I)^{1/2}f \right\rangle_{L_2(\mathrm{d}\rho)}$$

$$\geq 0.$$

since $(\Sigma + \lambda I)^{1/2}$ is self-adjoint. Similarly, we have $\Sigma - \hat{\Sigma} \succeq -t(\Sigma + \lambda I)$. This completes the proof. $\qquad \square$

Below, we provide the proofs of Theorem 3 by dividing it into items (i) and (ii).

**Theorem 7.** *Let $\delta \in (0, 1), \lambda > 0$ and $t > 0$. If*

$$M \geq \frac{4N_{q,\lambda}}{t^2} \log \frac{64N_{q,\lambda}}{\delta t^2},$$

*then, it holds*

$$\left\| K - \hat{K} \right\|_{L^2(\rho)} \leq \lambda \cdot t\delta^{-1} C_K^{(1)} + t^2 C_K^{(2)},$$

*with probability $1 - 2\delta$, where $C_K^{(1)}, C_K^{(2)}$ are constants depending on $K$.*

*Proof.* Using Lemma 6, we have

$$\left\|K - \hat{K}\right\|^2_{L^2(\rho\otimes\rho)} = \left\|\Sigma - \widehat{\Sigma}\right\|^2_{HS} = \operatorname{tr}\left[(\Sigma - \widehat{\Sigma})^2\right]$$

$$\leq t \cdot \operatorname{tr}\left[(\Sigma + \lambda I)(\Sigma - \widehat{\Sigma})\right]$$

$$= t \cdot \left(\operatorname{tr}\left[\Sigma(\Sigma - \widehat{\Sigma})\right] + \lambda \operatorname{tr}\left[\Sigma + \widehat{\Sigma}\right]\right)$$

$$\leq t \cdot \left(\operatorname{tr}\left[\Sigma\right]\left\|\Sigma - \widehat{\Sigma}\right\|_{\mathrm{op}} + \lambda \operatorname{tr}\left[\Sigma + \widehat{\Sigma}\right]\right),$$

with probability $1 - \delta$. Lemma 6 implies that $\left\|\Sigma - \widehat{\Sigma}\right\|_{\mathrm{op}} \leq t \cdot \|\Sigma + \lambda I\|_{\mathrm{op}}$, which yields

$$\left\|K - \hat{K}\right\|^2_{L^2(\rho\otimes\rho)} \leq t \cdot \left(t \cdot \operatorname{tr}\left[\Sigma\right]\|\Sigma + \lambda I\|_{\mathrm{op}} + \lambda \operatorname{tr}\left[\Sigma + \widehat{\Sigma}\right]\right)$$

$$\leq t \cdot \left(t \cdot \operatorname{tr}\left[\Sigma\right]\|\Sigma\|_{\mathrm{op}} + (1 + t)\lambda \operatorname{tr}\left[\Sigma\right] + \lambda \operatorname{tr}\left[\widehat{\Sigma}\right]\right)$$

$$\leq \lambda \cdot t\left(4\operatorname{tr}\left[\Sigma\right] + \operatorname{tr}\left[\widehat{\Sigma}\right]\right) + t^2 \cdot \operatorname{tr}\left[\Sigma\right]\|\Sigma\|_{\mathrm{op}}.$$

Now, $\operatorname{tr}[\Sigma]$ is bounded. Indeed, if the Mercer decomposition of $K$ is given by $K(x, y) = \sum_{j=1}^{\infty} \mu_j \psi_j(x)\psi_j(y)$, then we have

$$\operatorname{tr}[\Sigma] = \sum_{j=1}^{\infty} \mu_j = \int \left(\sum_{j=1}^{\infty} \mu_j \psi_j(x)\psi_j(x)\right)\mathrm{d}\rho(x) = \int K(x, x)\mathrm{d}\rho(x) < \infty,$$

since $x \mapsto K(x, x)$ is integrable with respect to $\rho$. Moreover, from the Markov inequality, we have

$$\mathbb{P}\left(\operatorname{tr}\left[\hat{\Sigma}\right] \geq \frac{\operatorname{tr}[\Sigma]}{\delta}\right) \leq \frac{\delta}{\operatorname{tr}[\Sigma]} \cdot \mathbb{E}_{z_1,\ldots,z_M}\left[\operatorname{tr}\hat{\Sigma}\right] = \frac{\delta}{\operatorname{tr}[\Sigma]} \cdot \operatorname{tr}[\Sigma] = \delta.$$

Therefore, it holds

$$\left\|K - \hat{K}\right\|^2_{L^2(\rho\otimes\rho)} \leq \lambda \cdot t\left(4 + \frac{1}{\delta}\right) \operatorname{tr}\left[\Sigma\right] + t^2 \cdot \operatorname{tr}\left[\Sigma\right]\|\Sigma\|_{\mathrm{op}} \leq \lambda \cdot t\delta^{-1}C_K^{(1)} + t^2 C_K^{(2)},$$

with probability $1 - 2\delta$, where $C_K^{(1)} = 5\operatorname{tr}[\Sigma]$ and $C_K^{(2)} = \operatorname{tr}[\Sigma]\|\Sigma\|_{\mathrm{op}}$. This completes the proof. $\square$

**Theorem 8.** *Let* $\delta \in (0, 1), \lambda > 0, t > 0$ *and* $\alpha \in [0, 1/2]$. *If*

$$M \geq \frac{4N_{q,\lambda}}{t^2} \log \frac{64N_{q,\lambda}}{\delta t^2},$$

*then, for any* $v : \mathbb{R}^d \to \mathbb{R}$ *with* $\|(\Sigma + \lambda I)^\alpha v\|_{L^2(\rho)} < \infty$, *it holds*

$$\left\|\int v(x)\hat{K}(x, \cdot)\mathrm{d}\rho(x) - \int v(x)K(x, \cdot)\mathrm{d}\rho(x)\right\|_{L^2(\rho)} \leq \sqrt{2}\left(C_{K,\alpha} + \lambda^{1-\alpha}\right) \cdot t\|(\Sigma + \lambda I)^\alpha v\|_{L^2(\rho)},$$

*with probability* $1 - \delta$, *where* $C_K^{(3)}$ *is a constant depending on* $K$.

*Proof.* The left-hand side can be bounded as

$$\left\|\int v(x)\hat{K}(x, \cdot)\mathrm{d}\rho(x) - \int v(x)K(x, \cdot)\mathrm{d}\rho(x)\right\|_{L^2(\rho)} = \left\|\hat{\Sigma}v - \Sigma v\right\|_{L^2(\rho)} = \left\|\left(\hat{\Sigma} - \Sigma\right)v\right\|_{L^2(\rho)}.$$

Since $\hat{\Sigma} - \Sigma$ is a self-adjoint operator, we have

$$\left\|\left(\hat{\Sigma} - \Sigma\right)v\right\|^2_{L^2(\rho)} = \left\langle v, \left(\hat{\Sigma} - \Sigma\right)^\dagger(\hat{\Sigma} - \Sigma)v\right\rangle$$

$$= \left\langle v, (\hat{\Sigma} - \Sigma)^2 v\right\rangle$$

$$= \left\langle (\Sigma + \lambda I)^\alpha v, \left((\Sigma + \lambda I)^{-\alpha}(\hat{\Sigma} - \Sigma)^2(\Sigma + \lambda I)^{-\alpha}\right)(\Sigma + \lambda I)^\alpha v\right\rangle$$

$$\leq \left\|(\Sigma + \lambda I)^{-\alpha}(\hat{\Sigma} - \Sigma)^2(\Sigma + \lambda I)^{-\alpha}\right\|_{\mathrm{op}}\|(\Sigma + \lambda I)^\alpha v\|^2_{L^2(\rho)}.$$

Now, let us bound $\left\|(\Sigma + \lambda I)^{-\alpha}(\hat{\Sigma} - \Sigma)^2(\Sigma + \lambda I)^{-\alpha}\right\|_{\mathrm{op}}$. Let $A = \Sigma - \hat{\Sigma}$ and $B = \Sigma + \lambda I$. Then, Lemma 6 implies that $\left\|B^{-1/2}AB^{-1/2}\right\|_{\mathrm{op}} \leq t$ with probability $1 - \delta$. Thus, we have

$$
\left\|(\Sigma + \lambda I)^{-\alpha}(\hat{\Sigma} - \Sigma)^2(\Sigma + \lambda I)^{-\alpha}\right\|_{\mathrm{op}}
$$
$$
= \left\|B^{-\alpha}A^2 B^{-\alpha}\right\|_{\mathrm{op}}
$$
$$
= \left\|B^{1/2-\alpha}\Big(B^{-1/2}A^2 B^{-1/2}\Big)B\Big(B^{-1/2}A^2 B^{-1/2}\Big)B^{1/2-\alpha}\right\|_{\mathrm{op}}
$$
$$
\leq \left\|B^{1/2-\alpha}\right\|_{\mathrm{op}}^2 \|B\|_{\mathrm{op}}\left\|B^{-1/2}A^2 B^{-1/2}\right\|_{\mathrm{op}}
$$
$$
\leq t^2 \|B\|_{\mathrm{op}}^{2(1-\alpha)},
$$

with probability $1 - \delta$. Moreover, since $2(1-\alpha) \geq 1$, we have

$$
\|B\|_{\mathrm{op}}^{2(1-\alpha)} = \|\Sigma + \lambda I\|_{\mathrm{op}}^{2(1-\alpha)} \leq \Big(\|\Sigma\|_{\mathrm{op}} + \lambda\Big)^{2(1-\alpha)}
$$
$$
\leq 2^{2(1-\alpha)-1}\Big(\|\Sigma\|_{\mathrm{op}}^{2(1-\alpha)} + \lambda^{2(1-\alpha)}\Big).
$$

Therefore, we conclude that

$$
\left\|\int v(x)\hat{K}(x,\cdot)\mathrm{d}\rho(x) - \int v(x)K(x,\cdot)\mathrm{d}\rho(x)\right\|_{L^2(\mathrm{d}\rho)}
$$
$$
\leq \left\|\Big(\hat{\Sigma} - \Sigma\Big)v\right\|_{L^2(\rho)}
$$
$$
\leq \sqrt{\left\|(\Sigma + \lambda I)^{-\alpha}(\hat{\Sigma} - \Sigma)^2(\Sigma + \lambda I)^{-\alpha}\right\|_{\mathrm{op}}}\|(\Sigma + \lambda I)^{\alpha}v\|_{L^2(\rho)}
$$
$$
\leq t\sqrt{\|B\|_{\mathrm{op}}^{2(1-\alpha)}}\|(\Sigma + \lambda I)^{\alpha}v\|_{L^2(\rho)}
$$
$$
\leq 2^{1/2-\alpha}\sqrt{\|\Sigma\|_{\mathrm{op}}^{2(1-\alpha)} + \lambda^{2(1-\alpha)}} \cdot t\|(\Sigma + \lambda I)^{\alpha}v\|_{L^2(\rho)}
$$
$$
\leq \sqrt{2}\Big(\|\Sigma\|_{\mathrm{op}}^{1-\alpha} + \lambda^{1-\alpha}\Big) \cdot t\|(\Sigma + \lambda I)^{\alpha}v\|_{L^2(\rho)}
$$
$$
\leq \sqrt{2}\Big(\max\{\|\Sigma\|_{\mathrm{op}}, \|\Sigma\|_{\mathrm{op}}^{1/2}\} + \lambda^{1-\alpha}\Big) \cdot t\|(\Sigma + \lambda I)^{\alpha}v\|_{L^2(\rho)}.
$$

This completes the proof. $\qquad\square$

## B   The Experimental Results on Next-Token Prediction

In this section, we report the performance of next-token prediction for the distilled models in Section 4. The results are presented in Table 7. Here, we highlight the following two key observations:

- Among the three types of loss, the layerwise losses sometimes underperform compared to the "direct" loss. This is natural because the cross entropy loss for next-token prediction is used in "direct", while $L^2$ loss (Softmax loss) just aims to make the attention kernel (attention weights) and its approximation close. As we reported in Section 4, for the downstream tasks, the models distilled with layerwise loss have comparable performance to the models distilled with "direct" loss.

- We observe that the models distilled using (layerwise) softmax loss outperform DiJiang, in which the learnable parameters are directly trained using next-token prediction loss. This result emphasizes the validity of our approach in training the features in linear attention.

## C   More Information of the Experiments

### C.1   Computational Resources and Hyperparameters

We provide the details of the experimental settings as follows:

Table 7: The performance of next-token prediction for the models distilled from GPT-2.

| | direct | softmax | $L^2$ | | Baseline method | Loss |
|---|---|---|---|---|---|---|
| Fix | 3.9657 | 5.4082 | 6.2453 | | DiJiang | 5.5965 |
| DoF | 5.2651 | 4.0170 | 6.6802 | | Performer | 8.1586 |
| DoF + Clip | 5.4547 | 4.0355 | 6.2378 | | Original GPT-2 | 3.3558 |

Table 8: Summary of learning rates.

| Model | PiQA | logiQA | ARC-E | ARC-C | Winogrande | MMLU | WSC |
|---|---|---|---|---|---|---|---|
| GPT-2 | 1e-4 | 5e-5 | 1e-4 | 1e-5 | 5e-4 | 1e-6 | 1e-6 |
| Pythia-1B | 1e-5 | 5e-5 | 5e-5 | 5e-5 | 5e-5 | 5e-6 | 5e-5 |

Table 9: Summary of batch sizes.

| Model | Distillation | PiQA | logiQA | ARC-E | ARC-C | Winogrande | MMLU | WSC |
|---|---|---|---|---|---|---|---|---|
| GPT-2 | 128 | 128 | 64 | 64 | 64 | 128 | 64 | 128 |
| Pythia-1B | 64 | 64 | 32 | 32 | 32 | 64 | 32 | 64 |

- Experiments for GPT-2 are conducted on four devices of A100 40GB. Experiments for Pythia-1B are conducted on four devices of A100 80GB.
- For Wikipedia dataset, we randomly sample 10% segment, and use it as one dataset.
- The context lengths for GPT-2 and Pythia-1B were set to 1024 and 2048, respectively.
- Training for distillation (learning features in proposed method / learning parameters in DiJiang) is conducted over 1 epoch. This consumes about 0.5 day for GPT-2 and 1 day for Pythia-1B.
- For the downstream tasks except for MMLU. training is conducted over three epochs for the tasks except for MMLU, and the best accuracy among three epochs is reported. For MMLU, we train the model for one epoch, and the accuracy for the last checkpoint is reported.
- As for the learning rates of distillation,
    - when using DiJiang, we set `0.02`.
    - when training feature maps, we set `0.02` for $z_1, \ldots, z_M$ and `0.2` for $\alpha_1, \ldots, \alpha_M$.
- The learning rates of downstream task are chosen to maximize the accuracy when we fine-tune the original model with softmax attention, and the same learning rates are used for the distilled model. The choices of learning rates are `1e-6, 5e-6, 1e-5, 5e-5, 1e-4, 5e-4` for GPT-2, and `1e-7, 5e-7, 1e-6, 5e-6, 1e-5, 5e-5` for Pythia-1B. The selected learning rates are summarized in Table 8.
- The batch sizes are summarized in Table 9. When out-of-memory error occurs, we utilized gradient accumulation.

## C.2  Datasets

All the datasets used in this paper are publicly available from HuggingFace Datasets library. The dataset URLs and licenses are summarized in Table 10.

Table 10: Summary of datasets.

| Dataset | URL | License |
|---|---|---|
| Wikipedia | `https://huggingface.co/datasets/legacy-datasets/wikipedia` | CC BY-SA 3.0 |
| PiQA | `https://huggingface.co/datasets/piqa` | AFL-3.0 |
| logiQA | `https://huggingface.co/datasets/EleutherAI/logiqa` | Apache License, Version 2.0 |
| ARC-E | `https://huggingface.co/datasets/allenai/ai2_arc` | CC BY-SA 4.0 |
| ARC-C | `https://huggingface.co/datasets/allenai/ai2_arc` | CC BY-SA 4.0 |
| Winogrande | `https://huggingface.co/datasets/allenai/winogrande` | Apache License, Version 2.0 |
| MMLU | `https://huggingface.co/datasets/mmlu` | MIT |
| WSC | `https://huggingface.co/datasets/wsc` | CC BY 4.0 |

## C.3  Additional Results

In this section, we provide the experimental results beyond Section 4.

Table 11: The approximation error when using $L^2$ loss for GPT-2. All values are expressed in units of $\times 10^{-3}$.

|  | 1 | 2 | 3 | 4 | 5 | 6 | 7 | 8 | 9 | 10 | 11 | 12 | Total |
|---|---|---|---|---|---|---|---|---|---|---|---|---|---|
| Fix | 8.70 | 10.60 | 3.74 | 2.39 | 3.48 | 1.77 | 1.80 | 1.21 | 1.19 | 1.22 | 1.32 | 8.26 | 45.68 |
| DoF | 7.83 | 8.92 | 4.15 | 2.50 | 3.48 | 1.76 | 1.76 | 1.24 | 1.33 | 1.29 | 1.32 | 8.52 | 44.10 |

Table 12: Training time (in hours and minutes) required for feature training.

| Model | Fix / DoF | direct | L2 | Softmax |
|---|---|---|---|---|
| GPT-2 | Fix | 10h 44m | 5h 57m | 8h 53m |
|  | DoF | 10h 45m | 6h 3m | 8h 59m |
|  | DoF + Clip | 10h 35m | 5h 43m | 8h 51m |
| Pythia-1B | Fix | – | – | 23h 30m |
|  | DoF | – | – | 24h 18m |

**Approximation error of $K$.**    Table 11 shows the loss values when GPT-2 is distilled into models whose dimensionality is determined by the Fix and DoF methods using the $L^2$ loss. Because the $L^2$ loss represents the error between $K$ and $\widehat{K}$, the values in this table directly indicate how well linear attention approximates softmax attention. From the table, we can see that the layers with small approximation errors under Fix also achieve small errors under the DoF. At the same time, for the layers with large approximation errors under the Fix method, the models whose dimensionality is selected by DoF exhibit smaller approximation errors. As a result, the total error across all layers is smaller for DoF than for Fix.

**Computational cost of Algorithm 1.**    The total time required to run Algorithm 1 is 105.05 seconds for GPT-2 and 476.53 seconds for Pythia-1B. These costs are negligible compared to the overall distillation process, which includes feature map training.

**Computational cost of feature training.**    We have summarized the actual training time required for feature training in Table 12. For GPT-2, the maximum training time is approximately 0.5 days, and for Pythia-1B, it is up to 1 day. Consistent with the trends shown in Table 5, the computational cost follows the order: $L^2 <$ softmax $<$ direct.

## C.4    Additional Discussion

**Why distilled models sometimes outperform the original model.**    Since the proposed method performs distillation by approximating the softmax attention with linear attention, the distilled models are generally expected to perform worse than the original models. However, as shown in Table 3 and Table 6, the distilled models outperform the original models in some tasks. Here, we discuss possible explanations for this phenomenon, in case readers are interested in understanding it.

One possible interpretation is that, the original model may not have reached the lowest possible test loss due to the limited amount of data available during downstream fine-tuning. Consequently, the error introduced by the approximate attention kernel in the distilled model might have incidentally contributed to a reduction in test loss. We believe that such a case occurred in our experiments.

Another possible explanation is that the original model may have slightly overfit the fine-tuning data for the downstream task, thereby limiting its test performance. In contrast, the distilled model might have benefited from an implicit regularization effect due to its smaller representational capacity, which could have led to improved generalization performance.

