# OpenReview forum: "Degrees of Freedom for Linear Attention: Distilling Softmax Attention with Optimal Feature Efficiency"
_NeurIPS.cc/2025/Conference — NeurIPS 2025 poster_

### Official Review · Reviewer_nGDS · 2025-07-01

**Clarity:** 3
**Significance:** 2
**Originality:** 2
**Rating:** 4
**Confidence:** 3

**Summary:**

The submission proposes a data-adaptive and layer-specific way to distill softmax attention in pre‑trained Transformers into linear attention via Kernel Approximation.

The core idea is to automatically choose the random feature dimension used in linear attention based on the statistical degrees of freedom (DoF) of the attention kernel. The authors derive a bound showing that the kernel approximation error depends on $N_{q, \lambda}$, a function of the sampling density, and show that sampling from a density proportional to  $\text{tr}(\Sigma(\Sigma+\lambda I)^{-1})$ (the DoF) yields an optimal sample complexity.

They then propose Algorithm 1 to estimate per‑layer DoF using a Gram matrix computed on sampled queries/keys from a pre‑trained model and allocate feature dimensions across layers under a budget constraint. To avoid expensive end‑to‑end distillation, the paper introduces layer‑wise training of random features via either an $L_2$ loss that matches attention kernels or a softmax (cross‑entropy) loss that matches attention distributions. Experiments distill GPT‑2 and Pythia‑1B models and evaluate on seven downstream tasks. Models using DoF‑based dimension selection and layer‑wise training match or exceed baselines with fixed dimensions (e.g., Performer and DiJiang) and sometimes approach the original model’s accuracy. The paper concludes with an analysis of effective dimensionality across layers and discusses limitations such as shared dimensions across heads.

**Questions:**

– To really unleash the benefits of linear attention, consider evaluating linear attention approaches on long-context tasks beyond standard benchmarks.

**Ethical Concerns:**

["NO or VERY MINOR ethics concerns only"]

**Final Justification:**

I would like to thank the authors for their rebuttal, and I will keep my original score.

**Quality:**

3

**Strengths And Weaknesses:**

**Strengths**:

- Clear motivation and problem formulation. The paper articulates the challenge of choosing a fixed random feature dimension for linear attention and motivates why different layers may require different dimensions. This is supported by theoretical analysis relating sample complexity to DoF.

- Principled algorithm with theoretical guarantees. The authors derive an error bound (Theorem 3) for approximating the softmax kernel by random features and show that sampling from the optimal density reduces the required number of features. Algorithm 1 provides a practical estimation of DoF and allocates per‑layer feature dimensions under a fixed cost.

**Weaknesses**:

- Limited evaluation scope. The experiments cover GPT‑2 (117M parameters) and Pythia‑1B (1B parameters), but there is no evaluation on larger models (e.g., GPT‑2 XL or LLaMA) or non‑language modalities. It’s unclear how the method scales to models with many more layers or to vision Transformers. The authors mention this limitation but do not provide preliminary evidence. It would be great if the authors could try their proposed approach on slightly larger models (such as LLaMA-3, Qwen, deepseek) and evaluate the models on a slightly broader tasks such as instruction following.

- Compared to the baseline (Fix), in Table 2 and Table 3, the proposed method is not significantly better than the simple baseline.

---

> ### Author Rebuttal · Authors · 2025-07-31
>
> We thank the reviewer for the insightful feedback. We address the specific concerns and questions below.
>
>
> > Limited evaluation scope. The experiments cover GPT‑2 (117M parameters) and Pythia‑1B (1B parameters), but there is no evaluation on larger models (e.g., GPT‑2 XL or LLaMA) or non‑language modalities. It’s unclear how the method scales to models with many more layers or to vision Transformers. The authors mention this limitation but do not provide preliminary evidence. It would be great if the authors could try their proposed approach on slightly larger models (such as LLaMA-3, Qwen, deepseek) and evaluate the models on a slightly broader tasks such as instruction following.
>
> We appreciate the reviewer’s insightful comment. While we acknowledge the benefits of evaluating on other modalities, larger models (e.g., LLaMA, Qwen, DeepSeek) and broader tasks (e.g., instruction following), we believe that the current experiments already provide strong evidence for the effectiveness and generality of our method.
>
> Our primary aim is to establish a theoretically grounded and practically effective framework for distilling softmax attention into linear attention with adaptive feature dimensions. To validate this, we chose two widely-used and open-source models that differ in scale. The results across both models consistently show that our dimension selection method improves performance. Importantly, we evaluated performance on a diverse set of downstream tasks, showing consistent improvements over strong baselines.
>
> We fully agree that extending our evaluation to larger models and broader tasks is a natural and valuable next step. Nonetheless, we believe our current empirical results, supported by rigorous theory, already demonstrate the core value and practicality of our method.
>
>
> > Compared to the baseline (Fix), in Table 2 and Table 3, the proposed method is not significantly better than the simple baseline.
>
> We thank the reviewer for raising this important point. We believe the current experimental results offer sufficient evidence for the effectiveness of our method over the Fix baseline, for the following reasons:
>
>
> First, our evaluation covers **multiple diverse downstream tasks**, and we report both task-wise scores and their average. This provides a comprehensive assessment of model quality. The proposed method achieves better average performance overall, which demonstrates its effectiveness under fixed resource constraints.
>
>
> Second, and most importantly, we believe the significance of our results lies not just in absolute accuracy improvements, but in **how much they help close the performance gap between linear attention and softmax attention**. In the context of distillation from a stronger teacher model, it is natural for a distilled model to underperform the original teacher model (while some exceptions exist in the experiments). Thus, the central goal is not to surpass the teacher, but to preserve as much of its performance as possible under tighter resource constraints. From this perspective, for instance, in Table 6 (Pythia-1B), the Fix baseline shows an average accuracy gap of 0.0265 compared to the original model. In contrast, our proposed DoF-based method reduces this gap to 0.0161, which implies **a 39% reduction in performance degradation**. We consider this a meaningful result, demonstrating that our method helps recover the performance typically lost when replacing softmax attention with linear attention.
>
>
> > To really unleash the benefits of linear attention, consider evaluating linear attention approaches on long-context tasks beyond standard benchmarks.
>
> Thank you for your thoughtful suggestion. We agree that evaluating linear attention models on long-context tasks is an important direction. However, our work focuses on distilling pre-trained Transformers, which are typically trained on inputs with a fixed context length. As a result, adapting the distilled model to operate on longer contexts than the original model is inherently challenging, and this difficulty is not specific to our proposed feature dimension selection method. Indeed, prior work such as Chen et al. (2024), which also distills Transformers into linear attention models, conducts evaluation only on standard language benchmarks.
>
> More importantly, the goal of our study is not to extend the usable context length of pre-trained Transformers through distillation. Rather, our aim is to construct more computationally efficient models that operate within the same context length, while minimizing performance degradation. We believe this goal is well supported by our current experimental results.
>
>
> We would be happy to clarify any concerns or answer any questions that may come up during the discussion period. We would greatly appreciate it if you could consider increasing the score once all concerns have been resolved.

---

> > ### Author Response · Authors · 2025-08-08
> >
> > As the discussion period is ending soon, we wanted to follow up to check whether our response has adequately addressed your concerns for an improved score. If there are any remaining questions or concerns, we would be happy to provide further clarification.
> >
> > Thank you for your time and consideration.

---

### Official Review · Reviewer_i3tm · 2025-07-02

**Clarity:** 3
**Significance:** 3
**Originality:** 3
**Rating:** 5
**Confidence:** 3

**Summary:**

The paper proposes a new distillation method to distill softmax attention models into linear attention models. The novelty lies in the dimension selection of the features used for distillation, which is computed via the effective degrees of freedom each layer (and head) needs.The paper also proposes two methods to learn the features from data and shows extensive empirical evidence to backup their claims.

**Questions:**

1. How computationally involved is the computation of the effective degrees of freedom (i.e., Algorithm 1)? You only report inference speed.
1. Are you setting the feature dimension of the heads to the largest value computed? This is not entirely clear in the paper, you only mention that you fix the dimension.
1. If yes (to the previous question), have you considered using the mean dimension across all heads in a layer (or any other suitable way)? This will reduce the performance of the head with the highest feature dimension but this might be mitigated by the other heads.
1. Do you have a hypotheses why the distilled models can outperform the original model? Is this mainly due to the learned features?

**Ethical Concerns:**

["NO or VERY MINOR ethics concerns only"]

**Final Justification:**

The authors adequately addressed my concerns in the rebuttal, therefore I will maintain my positive score. I think the motivation of the paper is good and in its revised form also suitable for presentation at NeurIPS.

**Limitations:**

The limitations are adequately addressed in a separate paragraph in the conlcusion.

**Paper Formatting Concerns:**

The appendix is not attached to the main text in the current form, but this can be rectified in the final submission.

**Quality:**

3

**Strengths And Weaknesses:**

**Strengths:**
- The proposed idea is novel and offers a new efficient way to distill softmax attention models into more efficient linear ones.
- The practical impact of the proposed method is potentially high and is bolstered by two new feature learning methods.
- The paper is well-written and easy to follow.

**Weaknesses:**
- The computational advantages are only sparingly shown in the empirical section. The computation times/performance of Algorithm 1 is not show nor discussed. Including this would give further information into how expensive the distillation process is. While there are some results for the achieved inference time after distillation, it would be beneficial to also show the inference time of the original softmax model to compare directly. Finally, more information on the computational complexity of the feature learning methods would be appreciated (apart from Table 5).
- The paper could benefit from a better setting in the current literature, i.e., the paper has relatively few citations. E.g. in Line 32, the references Wang et al. (2024) and Han et al. (2024) are not good references for the connection of linear attention to SSMs, better ones would be [1] and [2]. Additionally, mentioning more distillation methods would be nice, e.g., [3], [4].


**References:**

[1] Sieber et al., "Understanding the differences in foundation models: Attention, state space models, and recurrent neural networks", NeurIPS 2024

[2] Dao & Gu, "Transformers are SSMs: Generalized Models and Efficient Algorithms Through Structured State Space Duality", ICML 2024

[3] Yang et al., "Knowledge distillation via softmax regression representation learning", ICLR 2021

[4] Wang et al., "Minilm: Deep self-attention distillation for task-agnostic compression of pre-trained transformers", NeurIPS 2020

**Minor comments/Typos:**
- Please add the appendix to the main text for the final submission to improve the flow.
- Caption of Table 1: The underlined values are the three lowest values, not the dimensions lower than the top three.
- Add to captions of Table 4 and 5 that these values are for GPT-2.
- Line 11: ... smaller errors ...
- Line 34/35: Despite -> Even though; remove "indeed"

---

> ### Author Rebuttal · Authors · 2025-07-31
>
> We thank the reviewer for the helpful feedback. We address the specific concerns and questions below.
>
>
> > The computational advantages are only sparingly shown in the empirical section. The computation times/performance of Algorithm 1 is not show nor discussed. Including this would give further information into how expensive the distillation process is.
>
>
> > 1. How computationally involved is the computation of the effective degrees of freedom (i.e., Algorithm 1)? You only report inference speed.
>
>
> Thank you for your thoughtful suggestion. We have measured the runtime of Algorithm 1. The total time required is 105.05 seconds for GPT-2 and 476.53 seconds for Pythia-1B. These costs are negligible compared to the overall distillation process, which includes feature map training. We will include this information in the final version of the paper.
>
> > While there are some results for the achieved inference time after distillation, it would be beneficial to also show the inference time of the original softmax model to compare directly.
>
> We thank the reviewer for the valuable suggestion. We agree that comparing the inference time of the distilled linear attention model with the original softmax-based model could be informative. However, we did not include such a comparison in the paper for the following reasons:
>
> 1. **Implementation-dependence**: The inference time is highly implementation-specific. Modern Transformer libraries apply a wide range of kernel-level and architectural optimizations to softmax attention (e.g., FlashAttention), whereas our linear attention implementation does not yet incorporate such optimizations. As a result, a naive comparison of wall-clock time would reflect engineering disparities rather than algorithmic efficiency.
> 2. **Well-established fact**: The linear-time nature of linear attention compared to the quadratic complexity of softmax attention has been well documented in prior work. Our focus is not on re-establishing this fact, but rather on *how to optimally allocate feature dimensions* within a given linear attention budget to improve performance.
> 3. **DoF maintains the efficiency of Fix**: A central goal of our work is to enhance the performance of linear attention *without increasing its computational cost*. As shown in Table 4, our method adaptively allocates feature dimensions across layers while keeping the overall inference cost effectively unchanged. This confirms that the proposed DoF-based strategy improves accuracy while preserving the efficiency advantage of linear attention.
>
> > Finally, more information on the computational complexity of the feature learning methods would be appreciated (apart from Table 5).
>
> We have summarized the actual training time (in seconds) required for feature learning in the table below. We will include this table in the final version of the paper. For GPT-2, the maximum training time is approximately 0.5 days, and for Pythia-1B, it is up to 1 day. Consistent with the trends shown in Table 5, the computational cost follows the order: L2 < softmax < direct.
>
> | Model | Fix / DoF | direct | L2 | Softmax |
> | --- | --- | --- | --- | --- |
> | GPT-2 | Fix | 38659.7177 | 21420.2063 | 31991.2506 |
> |  | DoF | 38689.3816 | 21753.6426 | 32322.82 |
> |  | DoF + Clip | 38093.8039 | 20563.6596 | 31847.8818 |
> | Pythia-1B | Fix | - | - | 84601.079 |
> |  | DoF | - | - | 87506.1713 |
>
> > The paper could benefit from a better setting in the current literature, i.e., the paper has relatively few citations. E.g. in Line 32, the references Wang et al. (2024) and Han et al. (2024) are not good references for the connection of linear attention to SSMs, better ones would be [1] and [2]. Additionally, mentioning more distillation methods would be nice, e.g., [3], [4].
>
> Thank you for the important comment regarding related work. We agree that the current discussion of related work can be improved. We will revise Section 1 to include the more relevant references (Sieber et al., 2024; Dao & Gu, 2024) about the connection of linear attention to SSMs, as well as additional distillation methods (Yang et al., 2021; Wang et al., 2020).
>
> > **Minor comments/Typos:** …
>
> Thank you for pointing out these typos and for your helpful suggestions to improve the readability of the paper. We will incorporate all of these changes (including corrections to the captions, wording suggestions, and appendix integration) in the final camera-ready version.
>
> > Are you setting the feature dimension of the heads to the largest value computed? This is not entirely clear in the paper, you only mention that you fix the dimension.
>
> Thank you for pointing this out. Yes, we set the feature dimension for each layer to the maximum value among its heads. This is indeed mentioned in Algorithm 1 and also in line 207 of the paper, where we state: “Hence, we define the degrees of freedom …”. However, we understand that this point may not have been sufficiently emphasized. In the revised version, we will restate this point more clearly in experimental section (Section 4).
>
> > 3. If yes (to the previous question), have you considered using the mean dimension across all heads in a layer (or any other suitable way)? This will reduce the performance of the head with the highest feature dimension but this might be mitigated by the other heads.
>
> We are grateful for this suggestion. We have not explored using the mean feature dimension across all heads in a layer. Since our distillation procedure is conducted independently for each head, i.e., the loss functions defined in Section 3.3 are applied separately to each head, we believe it is unlikely that other heads can compensate for the reduced performance of a head whose feature dimension is insufficient. Therefore, lowering the feature dimension of a complex head would likely degrade its ability to approximate the attention kernel, which may in turn negatively affect the overall model performance.
>
>
> That said, we agree that this is an interesting direction, and exploring more flexible or shared allocation strategies across heads could be valuable for future work.
>
> > 4. Do you have a hypotheses why the distilled models can outperform the original model? Is this mainly due to the learned features?
>
> One explanation is that, in practice, the original model may not have reached the lowest possible test loss due to the limited amount of data available during downstream fine-tuning. As a result, the deviation caused by the approximated attention kernel in the distilled model may have incidentally reduced the test loss. We believe such a case occurred in our experiments.
>
> Another possible (though unverified) hypothesis is that the original model may have slightly overfit the fine-tuning training data, which limited its test performance. In contrast, the distilled models may have benefited from an implicit regularization effect since they have smaller representational capacity, thereby improving generalization.
>
> We would be happy to clarify any concerns or answer any questions that may come up during the discussion period.

---

> > ### Comment · Reviewer_i3tm · 2025-08-04
> >
> > Thank you for the detailed answers to my questions and suggestions.
> >
> > 1. I agree that the DoF computation times are negligible and I appreciate that you include the times in the revised manuscript.
> > 2. I also appreciate the inclusion of the table you provided in the answer. However, I would suggest to use hours (instead of seconds) in the manuscript, since this will be easier to parse for the reader. Also rounding to full seconds should be sufficient.
> > 3. Thank you for elaborating on the head-wise implementation and the performance of the distilled models. As a minor suggestion, I would asked the authors to think about including a sentence or two on the performance of the distilled models (essentially their answer to Q4) in the revised manuscript to provide additional insights for the reader.
> >
> > I don't have any additional concerns and I will maintain my positive score.

---

> > > ### Author Response · Authors · 2025-08-09
> > >
> > > We sincerely thank the reviewer for their positive feedback and constructive suggestions.
> > >
> > > We will change the unit of the times from seconds to hours, round the values to full seconds for clarity, and include the updated results in the revised paper. We will also add a brief description of the performance of the distilled models, summarizing our response to Q4.

---

### Official Review · Reviewer_uHta · 2025-07-02

**Clarity:** 3
**Significance:** 2
**Originality:** 2
**Rating:** 3
**Confidence:** 3

**Summary:**

This paper addresses the challenge of selecting the feature dimension when distilling pretrained softmax attention into more efficient linear attention models. The authors propose a method to determine the feature dimension for each layer based on the estimated degrees of freedom of the attention matrix. In addition, they advocate for layer-wise distillation rather than end-to-end training, arguing that it leads to better alignment and stability during knowledge transfer. The proposed approach is evaluated using GPT-2 and Pythia-1B models on the Wikipedia dataset, demonstrating promising results in terms of both efficiency and performance.

**Questions:**

1. The performance of baseline models (e.g., original, DiJiang, and Performer) in Tables 3 and 6 appears significantly lower than what has been publicly reported in previous works. Could the authors clarify why this is the case? Were different training setups, datasets, or hyperparameters used?
2. When comparing the proposed method with DiJiang and Performer in Table 3, efficiency metrics such as FLOPs or latency should be reported as well. Since the motivation is to distill softmax attention into linear attention while preserving efficiency, this comparison is incomplete without those figures.
3. In Table 3, it is unclear whether Fix should be considered a baseline or part of the proposed method. The central claim of the paper is that the DoF-based method can achieve better performance under the same feature dimension budget. However, in cases where Fix and DoF share the same budget, their performance is often quite similar. This undermines the claim that the DoF-based allocation provides a clear advantage. A more careful comparison and explanation are needed to clarify whether the observed gains are due to the DoF strategy itself or simply due to increased parameter capacity.
4. What does “DoF + clip” mean in Table 3? The phrase “clamp the feature dimensions to be at most the head size” is unclear—does it imply that feature dimensions can normally exceed the head size? If so, how is this handled in implementation, and why would such a configuration be valid?
5. Why are DiJiang, Performer, and “DoF + clip” missing from Table 6? Including these baselines would provide a more complete picture of the proposed method’s performance across settings.
6. The motivation for layerwise training is not clearly supported. In Table 3, the Direct method often achieves the best performance, raising the question of whether layerwise distillation offers any real advantage. Does layerwise training lead to faster convergence, better stability, or other benefits not captured by final performance? If not, its inclusion appears secondary and may not contribute meaningfully to the overall method.

Minor Suggestion:
- It is unclear why all attention heads are required to have the same feature dimension. Allowing different dimensions per head could potentially lead to better parameter efficiency and adaptivity. Have the authors considered or experimented with head-wise dimension selection?

**Ethical Concerns:**

["NO or VERY MINOR ethics concerns only"]

**Final Justification:**

This paper presents a good motivation. However, the proposed method appears overly complicated and lacks clarity, making it difficult to follow. Additionally, the results are not clearly presented or sufficiently convincing to support the claims. I believe the paper needs further polishing and clarification before it can be considered for acceptance.

**Limitations:**

yes

**Quality:**

1

**Strengths And Weaknesses:**

Strengths:
- The paper is clearly written and easy to follow with well-structured explanations.
- The idea of selecting the optimal feature dimension for each layer based on degrees of freedom is novel and supported by empirical evidence. The observation that different layers and attention heads exhibit significantly varying degrees of freedom suggests substantial redundancy in standard ViT-style architectures, making the analysis insightful and impactful.

Weaknesses:
- The theoretical derivation leading to the first and only formal result is overly long and contributes little to the main findings. Much of the content is a restatement or adaptation of existing results from prior work, rather than offering new theoretical insights directly tied to the proposed method.
- The proposal to use layer-wise training instead of end-to-end distillation lacks clear motivation and theoretical justification. It appears secondary to the main contribution and is not sufficiently developed.
- The experimental results, while promising, do not strongly support the claimed advantages of the proposed method. The gains are modest, and more thorough ablations or broader comparisons would be needed to convincingly demonstrate the effectiveness of the approach.

---

> ### Author Rebuttal · Authors · 2025-07-31
>
> We thank the reviewer for the helpful feedback. We address the specific concerns and questions below.
>
> > The theoretical derivation leading to the first and only formal result is overly long...
>
> We appreciate your important comment. We agree that parts of the theoretical section can be made more concise. However, we believe the current exposition plays an important role for two reasons.
>
> First, it introduces the key notation and concepts, such as the kernel approximation $\hat{K}$, the feature maps $\phi$, $\Phi$, and the operators $\Sigma$, $\hat{\Sigma}$, and $N_{q,\lambda}$, which are fundamental for our analysis and algorithm.
>
> Second, the derivation builds on prior work, and we include these results to **motivate and justify the use of degrees of freedom (DoF) as a core quantity for feature dimension selection**. This connection, though known in kernel theory, is not established in the context of linear attention, and we believe making it explicit clarifies the theoretical motivation behind our method.
>
> That said, we will revise and streamline parts of the exposition (e.g., lines 140–144 and around Proposition 1) to improve clarity in the final version. We appreciate your suggestion.
>
> > The proposal to use layer-wise training ...
>
> > 6. The motivation for layerwise training is not clearly supported.
>
> **The main motivation for introducing layer-wise training is to reduce the computational cost of distillation**, which is discussed in lines 52–55, 64–65, and 220–223. End-to-end training requires full backpropagation through all layers simultaneously, which leads to significantly higher training cost, especially for larger models. In contrast, layerwise training allows us to optimize each attention layer independently, avoiding high cost for computing gradient. Our experiments demonstrate that layerwise training is **not only more efficient** (Table 5) **but also yields performance comparable to end-to-end training** (Table 3, 6).
>
> > The experimental results, while promising, do not strongly support ...
>
> Thank you for your feedback. We would like to clarify that the experimental results do provide concrete evidence supporting the effectiveness of our proposed method.
>
> Our experimental results show that using our DoF-based feature dimension allocation leads to better downstream task performance compared to the “Fix”. Importantly, **Fix and DoF are matched in total feature dimension, and therefore incur the same computational cost**. This point may not have been sufficiently described, and **we will clarify it in Section 4.2 of the final version.** As further evidence, Table 4 confirms that the inference time for Fix and DoF is nearly identical. Thus, the observed performance improvement cannot be attributed to an increase in parameter capacity or computation budget, but rather to a more effective allocation of feature dimensions across layers.
>
> Crucially, in the context of distillation from a stronger teacher model, the key goal in distillation is minimizing performance drop from the teacher model. Therefore, when evaluating the benefits of our method over baseline “Fix”, the key metric is not the raw accuracy gain, but rather **how small the performance degradation from the original softmax-based model is**. From this perspective, the improvements are far from modest. For example, as shown in Table 6 (Pythia-1B), the Fix baseline exhibits a 0.0265 drop in average accuracy compared to the original model, whereas our DoF-based method reduces this gap to 0.0161, which implies **a 39% reduction in degradation**. We view this as a significant improvement given that the total inference cost remains unchanged.
>
> We agree that further ablations could strengthen the case, but we believe the current results already demonstrate the practical value of our approach.
>
> > 1. The performance of baseline models in Tables 3 and 6 ...
>
> We believe that the differences in baseline performance are mainly due to differences in experimental setup. In order to ensure a fair and controlled comparison, we fine-tuned all models under a unified setup. These settings may differ from those used in prior works, which often employ task-specific tuning. While such tuning can improve absolute performance, we did not apply it, as our goal is to demonstrate consistent relative improvements under equal resource constraints. Full details of our experimental setup are provided in Appendix C, and we also include our code in the supplementary materials to support reproducibility. The baseline implementations are taken from their official repositories.
>
> Although it is true that the downstream results for baseline models are sometimes lower than those reported in prior work our reproduced baselines are not consistently weaker: for example, when comparing results on DiJiang distilled by GPT-2 (117M) of our paper and results on DiJiang-160M by Chen et al. (2024), in some of the tasks, the former shows higher accuracy (e.g., logiQA, ARC-C, WSC).
>
> > 2. When comparing the proposed method with DiJiang and Performer in Table 3, ...
>
> Thank you for the insightful suggestion. Our goal is to improve downstream performance of linear attention under a fixed computational budget, rather than to further accelerate it. Accordingly, Table 3 compares accuracy at equal total feature dimensions, and Table 4 confirms that our method (DoF) maintains inference efficiency comparable to the fixed baseline (Fix).
>
> We agree that including efficiency metrics for methods like DiJiang would be valuable. However, DiJiang incorporates an acceleration technique using discrete cosine transforms which make it faster than standard linear attention. To emphasize the effect of adaptive dimension selection, we intentionally focused on a simple linear attention architecture without such modifications when comparing inference-time speed.
>
> Integrating our dimension selection method into more efficient architectures like DiJiang is a promising direction for future work.  We also emphasize that our DoF-based dimension selection is not in competition with specific linear attention mechanisms like DiJiang or Performer. Rather, it is complementary, and can be applied to various linear attention models, including DiJiang, as a general strategy for improving performance under a fixed budget.
>
> Finally, we note that Performer and Fix share the same architecture and compute cost; the difference lies in whether kernel features are learned or fixed.
>
> > 3. In Table 3, it is unclear whether Fix ...
>
> Thank you for this important question. We clarify that “Fix” serves as a baseline to evaluate the effect of DoF-based dimension allocation. Both Fix and DoF use trained feature maps under the same total feature dimension. In Fix, the feature dimension is uniformly set to the head size (as in prior work), while DoF allocates it per layer based on Table 1. Since the total dimensions of the both models are the same (see also line 246--248), the computational complexity of two models are the same. Thus, any performance difference reflects the impact of the allocation strategy itself, not differences in model capacity.
>
> Furthermore, we would like to emphasize again that DoF reduces the accuracy drop from the original model by 39% compared to Fix, which shows a substantial gain under matched budgets.
>
> We acknowledge that it was not sufficiently described that the feature dimension of “Fix” is set to be the same as the head size. We will clarify this point in the final version. Thank you again for the helpful question.
>
> > 4. What does “DoF + clip” ...
>
> DoF + clip is introduced in line 280--282. It refers to a variant of our method in which the feature dimensions computed via Algorithm 1 are clamped to be at most the head size. That is, for each layer, we set the feature dimension to $\min(\text{DoF-based value}, \text{head size})$, where DoF-based value corresponds to the values shown in Table 1.
> It is indeed possible in our method that the computed DoF-based feature dimension exceeds the head size (e.g., GPT-2 Layer 2 in Table 1 has a value of 182, while the head size is 64). This does not cause any implementation issue, as the feature dimension in linear attention can operate with arbitrary feature dimensions, regardless of the head size.
>
> > 5. Why are DiJiang, Performer, and “DoF + clip” missing from Table 6?
>
> Thank you for the suggestion. We excluded DiJiang, Performer, and DoF + Clip from the Pythia-1B results (Table 6) to focus on the key comparison between Fix and DoF, which directly evaluates our core contribution (i.e., feature dimension selection via Algorithm 1), under a larger model.
>
> These baselines were thoroughly evaluated in the GPT-2 setting (Table 3), where we showed that DoF outperforms both DiJiang and Performer. These results showed that using DoF for layer-wise dimension allocation and feature training yields improved downstream accuracy under the same feature dimensions. We chose not to repeat these comparisons in Table 6. We also note that DoF + Clip is an auxiliary variant for reducing inference cost and is not central to our main claim.
>
> > Minor Suggestion: It is unclear why all attention heads are ...
>
> We set the feature dimension to be the same across heads within a layer, as all heads are handled using a shared tensor (see line 205). This avoids overhead from irregular memory layouts or dynamic shapes.
>
> While head-wise allocation may offer further gains in adaptivity, it requires nontrivial architectural changes and scheduling complexity. We view this as an important direction for future work, but our current focus is on improving efficiency and accuracy within standard Transformer infrastructures.
>
> We would be happy to clarify any concerns or answer any questions that may come up during the discussion period. We would greatly appreciate it if you could consider increasing the score once all concerns have been resolved.

---

> > ### Comment · Reviewer_uHta · 2025-08-03
> >
> > Thank you for the clarification. I find the core motivation of the paper strong and interesting. However, the experimental results remain somewhat confusing. The performance comparisons lack clarity, and the ablation studies do not clearly isolate the source of improvement or convincingly validate the proposed motivation.
> >
> > Additionally, it is unclear whether the final proposed model corresponds to DoF or DoF-Clip. It seems that parts of the results support one variant while other parts contradict it, making it hard to assess the contribution coherently. Based on the available information, I have slightly increased my score to reflect the stronger motivation, but concerns remain regarding experimental rigor and clarity.

---

> ### Author Response · Authors · 2025-08-05
>
> Thank you very much for your thoughtful follow-up and for recognizing the strength and motivation behind our work. We truly appreciate your engagement and the time you have taken to review our paper and response.
>
> We sincerely appreciate your comment indicating that you slightly increased your score in light of the clarified motivation. However, we noticed that the score does not yet appear to have been updated in the system. If this was unintentional, we would be grateful if you could kindly update it to reflect your revised evaluation.
>
> We address the concerns raised by the reviewer as follows:
>
> > the experimental results remain somewhat confusing. The performance comparisons lack clarity, and the ablation studies do not clearly isolate the source of improvement or convincingly validate the proposed motivation.
>
> We would like to clarify the goals of our experimental design and highlight how our results support the core contributions of our work.
>
> The central contribution of our paper lies in the use of degrees of freedom (DoF) for feature dimension selection in linear attention. Accordingly, the primary goal of our experiments is to **empirically demonstrate the advantage of DoF-based dimension selection over the standard fixed-dimension approach (Fix)**. This point is clearly validated in our results. For example, when using the softmax loss during distillation, the performance degradation from the original attention is reduced by 75% for GPT-2 and by 39% for Pythia-1B when comparing DoF against Fix. These improvements substantiate the effectiveness of our dimension selection method.
>
> The second goal of our experiments is to **evaluate the effectiveness of our layerwise training strategy**. The results show that efficient layerwise training methods—especially softmax loss—achieve performance comparable to, or sometimes even better than, training with the pre-training loss (direct). While $L^2$ loss yields lower performance than the direct loss, it still outperforms prior baselines while offering notable training efficiency gains. This shows that our method enables practical and scalable distillation without sacrificing accuracy.
>
> Lastly, regarding baselines, we selected Performer and DiJiang, which are relevant and representative linear attention architectures. The results of comparisons with these methods further support the validity of our approach.
>
> > it is unclear whether the final proposed model corresponds to DoF or DoF-Clip. It seems that parts of the results support one variant while other parts contradict it, making it hard to assess the contribution coherently.
>
> In our work, both DoF and DoF-Clip are positioned as part of our proposed method. The DoF method directly reflects the theoretical foundation provided by Theorem 3, offering a principled way to select feature dimensions. In contrast, DoF-Clip is a practical modification that caps the feature dimension by the head size to reduce computational cost.
>
> Importantly, our goal is not to emphasize the comparison between DoF and DoF-Clip, but rather to show that **both methods are effective compared to the fixed-dimension baseline (Fix)**. As shown in Table 3 (and Table 6 for DoF), both variants lead to better performance than Fix, demonstrating the effectiveness of DoF-based dimension selection (including DoF-Clip). This core contribution remains consistent across our experiments.
>
> While we did not intend to directly compare DoF and DoF-Clip as competing methods, we observed that DoF-Clip sometimes achieved higher performance than DoF. One possible explanation is that the reduced capacity of DoF-Clip may help prevent overfitting.
>
> We thank you again for your constructive feedback.

---

> > ### Comment · Reviewer_uHta · 2025-08-07
> >
> > Thank you for the clarification. I have no further concerns and will adjust my score accordingly to reflect the strengths of the paper.

---

### Official Review · Reviewer_RVhi · 2025-07-09

**Clarity:** 3
**Significance:** 3
**Originality:** 3
**Rating:** 5
**Confidence:** 4

**Summary:**

In this work, the authors study an interesting practical problem in sequence modeling: Distilling Softmax Attention into Linear Attention. The authors conduct a series of theoretical analyses on the selection of optimal feature dimensions and propose a layer-wise training method. Empirical verifications of the proposed algorithms are provided.

**Questions:**

1. Comparing fix to DoF, if we fix the feature dimension in every layer and head to be the max of all optimal dimensions, will the fix approach be better than the DoF approach? If not, could the authors explain more about it?
2. The authors could add discussions on [1, 2], which also provide analyses on the approximation quality of PRF-based Linear Attention and Softmax Attention in different perspectives, e.g., Theorem 3,4 in [1] and Theorem 3 in [2].

[1] Rethinking Attention with Performers. ICLR 2021.

[2] Stable, Fast and Accurate: Kernelized Attention with Relative Positional Encoding. NeurIPS 2021.

**Ethical Concerns:**

["NO or VERY MINOR ethics concerns only"]

**Limitations:**

The authors carefully discuss limitations and future work.

**Quality:**

3

**Strengths And Weaknesses:**

## Strengths
- The studied problem is interesting and significant, which helps to improve the utility of linear attention in practice.
- The proposed algorithms are well-motivated and supported by a series of interesting theoretical analyses.
- The paper is well-structured and easy to follow. The authors explain well after each theoretical statement to help readers catch up quickly.

## Weaknesses
- **Lack of empirical investigations on the approximation error when using the proposed DOF (and Clip)**. It would be better for the readers to understand the effectiveness of the proposed approach to further investigate the approximation error between Linear Attention and Softmax Attention in practice. Although the authors provide an analysis of it in Theorem 3, the readers would be curious about the empirical comparisons, especially its relations to the number of feature dimensions.
- **Limited scale of the experiments**. We can see the effectiveness of the proposed selection methods on GPT-2/Pythia-1B, but both are small-scale experiments. It is understandable that the computational resources would be a restriction for larger-scale experiments, but it would be better if the effectiveness of the approach could be verified further.
- **Limited choices of the kernel feature map**. In this work, the authors only investigate the PRF empirically. What is the effect of the proposed approach on other types of kernel feature maps? Will it be consistent adaptable or not? Further investigations could enhance the quality of this submission.

---

> ### Author Rebuttal · Authors · 2025-07-31
>
> We thank the reviewer for the helpful feedback. We address the specific concerns and questions below.
>
> > **Lack of empirical investigations on the approximation error when using the proposed DOF (and Clip)**. It would be better for the readers to understand the effectiveness of the proposed approach to further investigate the approximation error between Linear Attention and Softmax Attention in practice. Although the authors provide an analysis of it in Theorem 3, the readers would be curious about the empirical comparisons, especially its relations to the number of feature dimensions.
>
> Thank you for the insightful suggestion. In general, increasing the number of feature dimensions reduces the approximation error between linear and softmax attention, while decreasing it increases the error. Although our dimension selection strategy is theoretically justified, it is challenging to directly evaluate the effectiveness of the selected dimensions purely from an approximation error standpoint.
>
> In this work, following common practice in language model distillation, we chose to evaluate our method based on downstream task performance, which reflects the overall effectiveness in practical scenarios. The improved results under our DoF-based selection empirically validate that the approximation quality is improved when dimensions are allocated adaptively.
>
> As a reference, we provide below an empirical analysis from our GPT-2 experiments, showing the selected feature dimensions per layer and the change in approximation error (DoF/Fix ratio) before and after applying DoF-based selection. Here, the Fix baseline used a uniform dimension of 64. In layers where DoF selected larger dimensions than Fix, the approximation error decreased. On the other hand, in layers where smaller dimensions were selected, the error increased. This behavior is consistent with theoretical expectations. More importantly, the overall downstream performance improved, confirming that DoF-based allocation offers a better tradeoff between capacity and approximation quality.
>
> |      | 1    | 2    | 3   | 4   | 5   | 6   | 7   | 8   | 9   | 10  | 11  | 12  |
> |------|------|------|-----|-----|-----|-----|-----|-----|-----|-----|-----|-----|
> | Dimension | **130** | **182** | 35  | 44  | 42  | **65** | **92** | 33  | 28  | 34  | 46  | 39  |
> | DoF/Fix | 0.900 | 0.838 | 1.111 | 1.047 | 1.000 | 0.995 | 0.981 | 1.023 | 1.114 | 1.051 | 1.004 | 1.031 |
>
> > **Limited scale of the experiments**. We can see the effectiveness of the proposed selection methods on GPT-2/Pythia-1B, but both are small-scale experiments. It is understandable that the computational resources would be a restriction for larger-scale experiments, but it would be better if the effectiveness of the approach could be verified further.
>
> We thank the reviewer for raising this important point. While we acknowledge the benefits of evaluating on larger models, we believe the current experiments already provide strong evidence for the effectiveness and generality of our method.
>
> Our primary aim is to establish a theoretically grounded and practically effective framework for distilling softmax attention into linear attention with adaptive feature dimensions. To validate this, we deliberately chose two widely-used and open-source models that differ in scale. Importantly, we evaluated performance on a diverse set of downstream tasks, showing consistent improvements over strong baselines.
>
> We fully agree that extending to larger models such as LLaMA is valuable, and view it as promising directions for future work.  Nonetheless, we believe our current empirical results, supported by rigorous theory, already demonstrate the core value and practicality of our method.
>
> > **Limited choices of the kernel feature map**. In this work, the authors only investigate the PRF empirically. What is the effect of the proposed approach on other types of kernel feature maps? Will it be consistent adaptable or not? Further investigations could enhance the quality of this submission.
>
> Our theoretical framework is designed to apply to feature maps $\phi$ that satisfy the condition $K(x, y) = \mathbb{E}_{z \sim \tau}[\phi(x; z)\phi(y; z)]$, i.e., where the attention kernel can be expressed as an expectation over random features. Consequently, our proposed approach is not applicable to methods such as those in Katharopoulos et al. (2020) and Qin et al. (2022), which do not involve in randomized feature expectations.
>
> Among approximations of attention kernel that satisfy the above condition, PRF are the most well-established and promising, which motivated our focus on them in this work. Another common alternative is the use of random fourier features as in Peng et al. (2021). However, as pointed out in Choromanski et al. (2021), this can yield negative values, which may adversely affect stability when training. Since such implementation aspects beyond feature dimension can significantly influence performance, we chose not to adopt RFF in our study. This is briefly discussed in lines 127--128 of our paper.
>
> That said, our method is general, and if a promising alternative feature map for approximating an attention kernel $K$ (satisfying the expectation condition above) is developed, our dimensionality selection approach can readily be extended to it. We believe that including this discussion in the camera-ready version will make the paper more convincing. We appreciate the helpful comment.
>
>
> > 1. Comparing fix to DoF, if we fix the feature dimension in every layer and head to be the max of all optimal dimensions, will the fix approach be better than the DoF approach? If not, could the authors explain more about it?
>
> Thank you for the interesting question. We conducted additional experiments on GPT-2, where we fixed the feature dimension of all layers and heads to the global maximum value observed in Table 1 (i.e., 182). The results are as follows:
>
> Using the softmax loss, the average downstream accuracy across all tasks was 0.4149, with individual task scores of: 0.5724 (PiQA), 0.2842 (logiQA), 0.3232 (ARC-E), 0.2875 (ARC-C), 0.5004 (Winogrande), 0.3007 (MMLU), and 0.6346 (WSC). These are very close to the performance achieved by our DoF-based method. One possible reason is that the approximation of the layer with the largest DoF became a bottleneck, and increasing the feature dimensions of other layers did not improve the overall approximation quality of the network.
>
> Using the direct loss (i.e., next-token prediction loss), the average performance dropped to 0.4149, with task-wise scores: 0.5626, 0.2826, 0.2997, 0.2875, 0.5020, 0.2735, and 0.6442. This is lower than the performance achieved by the “DoF” in Table 3. We hypothesize that this drop may be due to the more complex optimization landscape of the direct loss, which could make it harder and unstable to learn the feature maps. For reference, when we fixed all feature dimensions to 128 and used the direct loss, the average accuracy was 0.4190, which is higher than the 182-fixed version, and close to the DoF result.
>
> > 2. The authors could add discussions on [1, 2], which also provide analyses on the approximation quality of PRF-based Linear Attention and Softmax Attention in different perspectives, e.g., Theorem 3,4 in [1] and Theorem 3 in [2].
>
> Thank you for the valuable comment to connect our theory to prior results. We summarize the relationship to [1, 2] as follows. We will add these discussions to the final version of the paper, and include [2] as an additional citation, which was previously omitted.
>
> Theorem 3 and 4 in [1] analyze the approximation quality of PRF-based linear attention in terms of the **sup-norm error** between the true attention kernel and its approximation. They show that to achieve an error less than $\epsilon$, the required number of features $M$ depends on the head dimension $d$, the norm bound $R$ of queries and keys, and $\epsilon$, with a particular dependence of $M = O(d \log d)$ on $d$. In contrast, our Theorem 3 focuses on bounding the **$L^2(\rho \otimes \rho)$-norm**, where $\rho$ represents the data distribution over queries and keys. This allows us to capture the structure of the input distribution in each attention layer. As a result, the approximation error is governed not by the dimension $d$, but by the degrees of freedom (DoF), which reflect the intrinsic dimensionality of the data.
>
> Theorem 3 in [2] highlights an exponential dependence on the norm bound $R$ of queries and keys. In our analysis, such dependence is implicitly captured by constants such as $C_K$ and $||K||^2_{L^2(\rho \otimes \rho)}$, which may also grow exponentially with $R$. Our work does not aim to reduce the exponential dependence on $R$; instead, we focus on how to select the feature dimension per layer to minimize the approximation error under a fixed computational budget.
>
> It is also worth noting that [2] proposes mitigating the exponential dependence on $R$ by normalizing queries and keys and incorporating relative positional encodings. While this is a promising direction, our setting assumes distillation from pre-trained models, where such modifications to the architecture or inputs are not directly applicable. That said, developing distillation techniques that incorporate such normalization to reduce sensitivity to $R$ remains an important and interesting direction for future work. We will mention this in the revised paper.
>
> We would be happy to clarify any concerns or answer any questions that may come up during the discussion period.

---

> > ### Author Response · Authors · 2025-08-08
> >
> > Thank you very much for your positive evaluation and constructive comments on our paper.
> >
> > As the discussion period is ending soon, we wanted to follow up to check whether our response has adequately addressed your concerns. If there are any remaining questions or concerns, we would be happy to provide further clarification.
> >
> > Thank you again for your time and consideration.

---

### Comment · Area_Chair_GedJ · 2025-08-06

Dear Reviewer,

Please review the authors’ rebuttal and finalize your scores. Please write explanations about your updated (or not-updated) scores and submit the Mandatory Acknowledgement.

Your effort is greatly appreciated for the conference.
Thanks, AC

---

### Note · Authors · 2025-08-16

We sincerely thank all reviewers and the Area Chair for their time and valuable feedback. We also appreciate that the reviewers recognized the significance of our method to select the dimension of linear attention for each layer using a theoretically supported approach.

Based on the reviewers’ feedback, we will incorporate the following revisions and improvements in the final version:

- **Clarifying the theoretical explanation**: We will emphasize that our theory applies to any feature map that satisfies the condition stated on line 111. We will also simplify some parts of the explanation and provide a more detailed discussion of the insights derived from our main result.
- **Addressing concerns about the strength of experimental support**: We will clarify how our results show that dimension selection based on DoF outperforms "Fix" (i.e., using the same dimension for all layers), responding to comments that the results may not strongly support our method. As we consider distillation, the goal is to minimize performance drop from the original models. Our results demonstrate that, for example, when using the softmax loss, performance degradation from the original model is reduced by 75% for GPT-2 and by 39% for Pythia-1B when comparing DoF against Fix.
- **Strengthening the experimental setup description**: We will explicitly state that, in the "Fix" setting, the dimension of each layer equals the head size, which was not previously mentioned. We will also reiterate, in Section 4, that the per-layer dimensions in our method are chosen based on the maximum degrees of freedom among the heads (previously mentioned only on line 207).
- **Adding more detailed experimental results**: We will include (i) the approximation error of the distilled linear attention, (ii) the computation time of Algorithm 1, and (iii) more information on computation time for feature training beyond Table 5. Furthermore, we will provide an explanation for cases where the distilled model outperforms the original model.
- **Expanding references**: We will include the additional references suggested by the reviewers on (i) the theoretical approximation capability of linear attention, (ii) the connection between linear attention and SSM-based models, and (iii) distillation. We will also add a discussion on the relationship between our theory and previous theoretical work.
- **Other corrections**: We will fix typos pointed out by the reviewers and add missing information to figure captions.

---

### Decision · Program_Chairs · 2025-09-17

**Decision:**

Accept (poster)

**Comment:**

This paper proposes a data/layer-specific method to distill softmax attention into linear attention via Kernel Approximation, where the key is to select the optimal feature dimensions based on degree of freedom via a series of theoretical analyses. Empirical verifications of the proposed algorithms are provided.

The reviewers in general agree that the work proposes an interesting and solid method to distill softmax attention models into more efficient linear ones. In the discussions, the authors further addressed some concerns from reviewers, eg, more detailed experimental results. Please incorporate these improvements into the final version if accepted.